# Population-level impacts of antibiotic usage on the human gut microbiome

Kihyun Lee ⬤[1,2], Sebastien Raguideau[3], Kimmo Sirén[4], Francesco Asnicar ⬤[5], Fabio Cumbo ⬤[5], Falk Hildebrand[3,6], Nicola Segata ⬤[5], Chang-Jun Cha ⬤[1,8] ✉ & Christopher Quince ⬤[3,6,7,8] ✉

The widespread usage of antimicrobials has driven the evolution of resistance in pathogenic microbes, both increased prevalence of antimicrobial resistance genes (ARGs) and their spread across species by horizontal gene transfer (HGT). However, the impact on the wider community of commensal microbes associated with the human body, the microbiome, is less well understood. Small-scale studies have determined the transient impacts of antibiotic consumption but we conduct an extensive survey of ARGs in 8972 metagenomes to determine the population-level impacts. Focusing on 3096 gut microbiomes from healthy individuals not taking antibiotics we demonstrate highly significant correlations between both the total ARG abundance and diversity and per capita antibiotic usage rates across ten countries spanning three continents. Samples from China were notable outliers. We use a collection of 154,723 human-associated metagenome assembled genomes (MAGs) to link these ARGs to taxa and detect HGT. This reveals that the correlations in ARG abundance are driven by multi-species mobile ARGs shared between pathogens and commensals, within a highly connected central component of the network of MAGs and ARGs. We also observe that individual human gut ARG profiles cluster into two types or resistotypes. The less frequent resistotype has higher overall ARG abundance, is associated with certain classes of resistance, and is linked to species-specific genes in the *Proteobacteria* on the periphery of the ARG network.

The acquisition of antimicrobial resistance (AMR) by human pathogens is well-established as one of the most serious current and developing threats to human health. It is estimated that over 30,000 deaths in Europe in 2015 were attributable to AMR infections[1], and this impact is growing[2]. To date, the majority of AMR surveillance consists of resistance rates in pathogen isolates cultured from samples taken from infected individuals. However, the majority of organisms that live on or in the human body are not pathogens but commensal components of the human microbiome. Antibiotic usage will impose a selective pressure, not just on the target pathogens, but the whole microbiome[3], and given that many antibiotic resistance genes (ARGs) are found on mobile genetic elements (MGEs) and are therefore frequently horizontally transmitted[4,5], it is vital for us to study not just AMR in pathogens, but also the wider impact of antibiotics on the aggregate

[1]Department of Systems Biotechnology and Center for Antibiotic Resistome, Chung-Ang University, Anseong 17546, Republic of Korea. [2]CJ Bioscience, Seoul 04527, Republic of Korea. [3]Organisms and Ecosystems, Earlham Institute, Norwich NR4 7UZ, UK. [4]Section for Evolutionary Genomics, The GLOBE Institute, University of Copenhagen, Copenhagen, Denmark. [5]Department of Cellular, Computational and Integrative Biology, University of Trento, Trento, Italy. [6]Gut Microbes and Health, Quadram Institute, Norwich NR4 7UQ, UK. [7]Warwick Medical School, University of Warwick, Coventry CV4 7HL, UK. [8]These authors jointly supervised this work: Chang-Jun Cha, Christopher Quince. ✉e-mail: cjcha@cau.ac.kr; christopher.quince@earlham.ac.uk

collection of resistance genes, or resistome[6], of the commensal microbiota. In particular, the most numerically abundant component of the human microbiota, the gut microbiome[7], has the potential to be an important reservoir of AMR[8].

There is now a significant body of research focussed on the transient impact of antibiotics on individual human gut microbiomes. These studies typically involve a relatively small number of individuals and/or follow their subjects for a limited amount of time. The first studies used 16S rRNA gene sequencing to follow changes in the microbial community structure associated with antibiotic treatment[9]. Substantial inter-individual variability was observed in the gut microbiome response to antibiotics, but in many cases, an increased relative abundance of *Enterobacteriaceae* and other potential pathogens was observed, with a concomitant reduction in more beneficial commensal organisms such as butyrate producers and a reduction in species diversity[10].

More recently shotgun metagenomics has enabled the impact of antibiotics on both the overall functional gene content of the community and variants within species to be determined. This has confirmed the substantial but mostly transient changes in community structure but not always with clear associations between the abundance of specific resistance genes and the antibiotic used[10–13]. Metagenomics combined with recently developed Hi-C library preparation strategies that allow a higher proportion of mobile ARGs to be associated with taxa have demonstrated that ARGs can transfer horizontally between gut microbes during the course of antibiotic treatment[14].

These studies demonstrate the significant transient impact that antibiotics can have on the human microbiota but the consequences of this impact at a population scale and over longer time periods are still relatively under-explored. There are clear geographic differences in the frequency and type of resistance observed in clinical isolates and these are associated with patterns of antibiotic usage[15]. There is also direct recent evidence from longitudinal studies on travellers to countries with high levels of resistance that resistant strains can be acquired even in the absence of treatment with antibiotics and then persist in the microbiome[16–18].

Clearly, the human microbiome is not an isolated system, strains are transmitted between hosts, and horizontal gene transfer (HGT) will occur between strains within hosts and in the environment. The result is that we can imagine an individual microbiome as receiving a constant stream of immigrant strains from a metapopulation[19] and with mobile genes in those strains being sampled from a mobilome that is shared at least amongst related species[20]. It is likely therefore that the widespread use of antibiotics will lead to ARGs not just increasing in abundance amongst individuals that are directly exposed but becoming endemic throughout the population, and this resistance then spreading throughout the microbiota.

There is already evidence of this, Forslund et al. (2013) demonstrated clear differences in ARG profiles between three countries for which deeply sequenced metagenome data sets were available at that point[21], and for four European countries a correlation between antibiotic resistance potential and outpatient antibiotic consumption, albeit at shallower sequencing depths. These differences between countries and also anomalously high resistance levels in China, have been confirmed by more recent studies[22], sometimes using as many as a thousand samples[23], but an explicit correlation between antibiotic consumption and resistance at a global scale has yet to be unequivocally demonstrated[24]. There is also the question of what is driving these geographical differences in the gut resistome, are the ARGs pathogen associated or commensal, are they species specific or mobilised.

We take advantage of the recent availability of both large-scale curated human microbiome metagenome data sets and genome binning of these same samples into metagenome assembled genomes (MAGs)[25,26] to perform a comprehensive population-level study of ARGs in the human gut microbiome. We build on previous analyses both in terms of scale and by more carefully attributing the response to antibiotic consumption to different types of ARGs, species-specific or mobile, and by cataloguing the mechanisms which are mobilising them[21–23].

We use both an assembly-based analysis strategy[27,28], identifying open reading frames prior to annotation of the entire gene, and a read-based method, mapping reads to the CARD, an AMR gene database[29], without assembly. The former, enabled us to compare AMR ORFs to a large-scale genome collection, over 300,000 genomes, derived approximately equally from human microbiome MAGs and reference isolate genomes, and detect mobile ARGs as those shared by different species. The latter, was found to be more sensitive for the family-level ARG profiling of individual samples. The identification of mobile genes as assembled sequence shared across taxa was complemented by direct screening of sequences corresponding to different types of MGEs including a machine learning method to identify plasmid contigs and thereby the ARGs on those plasmids[30]. The results of these analyses were then compared with information on population-level antibiotic usage to reveal the impact of antibiotic consumption on resistance in the human gut microbiome at a population-scale. This approach also enabled us to construct a bipartite network of ARGs and microbiome species.

Using this methodology, we show that (i) the per capita antibiotic consumption rate in a country correlates with the abundance of resistance genes in the population, (ii) such correlation is principally driven by mobile resistance genes embedded in a central network component dominated by commensal organisms, (iii) two distinct types of human gut resistome (resistotypes) exist.

## Results
### A comprehensive catalogue of ARGs from the human microbiome

We created a catalogue of ARGs across both the human microbiome and reference genomes by locating open reading frames (ORFs) on metagenomic assemblies from 8972 human microbiome samples spanning gut (7589), oral cavity (746), skin (380), airway (118), nasal cavity (55), and vagina (83) − (sample details are given in Table S1 with metadata included in Supplementary Data 1). More specifically, sample types classified as 'oral cavity' include samples from plaque (222), tongue (189), buccal mucosa (118), others or unspecified (217); 'nasal cavity' includes anterior nares (55); 'airway' corresponds to sputum (118). These human microbiome ORFs were combined with the ORFs identified on 151,655 bacterial and 842 archaeal genomes obtained from NCBI RefSeq. These genomes included representatives from the principal phyla found in the human gut microbiome although they were dominated by *Proteobacteria* (54.7%). ORFs across all these data were then annotated to ARGs at a stringent 80% amino acid identity across at least 80% of the target sequence[31], using a custom version of the Comprehensive Antibiotic Resistance Database − CARD ([28] and see Methods). We identified a median of 15 ARGs per metagenome across all body sites with a maximum of 242 and retrieved a total of 216,849 ARGs from the human microbiome. There were slightly more ARGs recovered per sample from airways (median 17.5) and stool (median 16) than from oral cavity (median=11) and skin (median= 4). Note that ARG diversity comparisons need to account for sampling depth as discussed below.

These microbiome ARGs were then pooled together with the 2,349,728 ARGs annotated from the RefSeq genomes and clustered at multiple levels (90%, 95%, 99%, 100%) of nucleotide identity (see Fig. 1 and Methods). These cut-offs were chosen to give varying levels of taxonomic resolution with 95% roughly corresponding to species[32]. In total, we identified 65,260 unique ARG sequences from the human microbiome samples, and these sequences represented 15907 gene variants at 99% nucleotide identity (Table S2). We will denote them as

'ARG_cluster99' for convenience, and the clusters generated at other cut-offs as ARG_clusterX (where X is the percent identity cutoff; see Table S2). Of the ARG_cluster99 that occurred in the microbiome samples, 60.9% were not found in the reference genome collection, for the ARG_cluster90 this was true of 18.0%, indicating that we have uncovered substantial previously unknown ARG diversity directly from human-associated metagenomes even at relatively large sequence divergence (Table S2).

## ARG novelty varied with respect to antibiotic classes

The degree of ARG novelty compared to the isolate genomes varied across ARG families and the antibiotic class that they provide resistance to (Fig. S1). Sulfonamides and peptide antibiotics were the two antibiotic classes with the lowest proportion of ARG_cluster99s exclusively composed of metagenomic ORFs (42.9% out of 21 clusters, 44.0% out of 1522 clusters, respectively). Tetracyclines and amphenicols, on the other hand, were associated with the highest proportion of uniquely metagenomic ARG_cluster99s (85.1% out of 1,656 clusters, 85.7% out of 342 clusters, respectively). Of the ARG families for which

at least 100 ARG_cluster99s were discovered in the microbiome samples, two tetracycline resistance genes *tetA(60)* (99.0% out of 100 clusters) and *tetB(60)* (97.5% out of 122 clusters) displayed the highest proportion of uniquely metagenomic clusters. Two multi-drug resistance genes, *msbA* (35.4% out of 144 clusters) and *acrF* (32.4% out of 145 clusters) were found to have the lowest proportion of uniquely metagenomic clusters.

## ARG diversity varied across body sites

The gut microbiome contributed the majority of the metagenome-unique ARG_cluster99 that were uncatalogued in the RefSeq genomes (6810 ARG_cluster99s, 70.3% of the metagenome-unique clusters). However, gut microbiome samples were over-represented in our data set, so this may give a false impression of their true relative contribution to ARG diversity. To address this we performed rarefaction, calculating ARG_cluster99 numbers in random subsets of a hundred samples (see Methods). Using this method, the gut had the highest diversity per sequenced sample with a median ARG_cluster99 richness of 9.3, compared to oral cavity with 6.2, airways with 5.6 and skin with

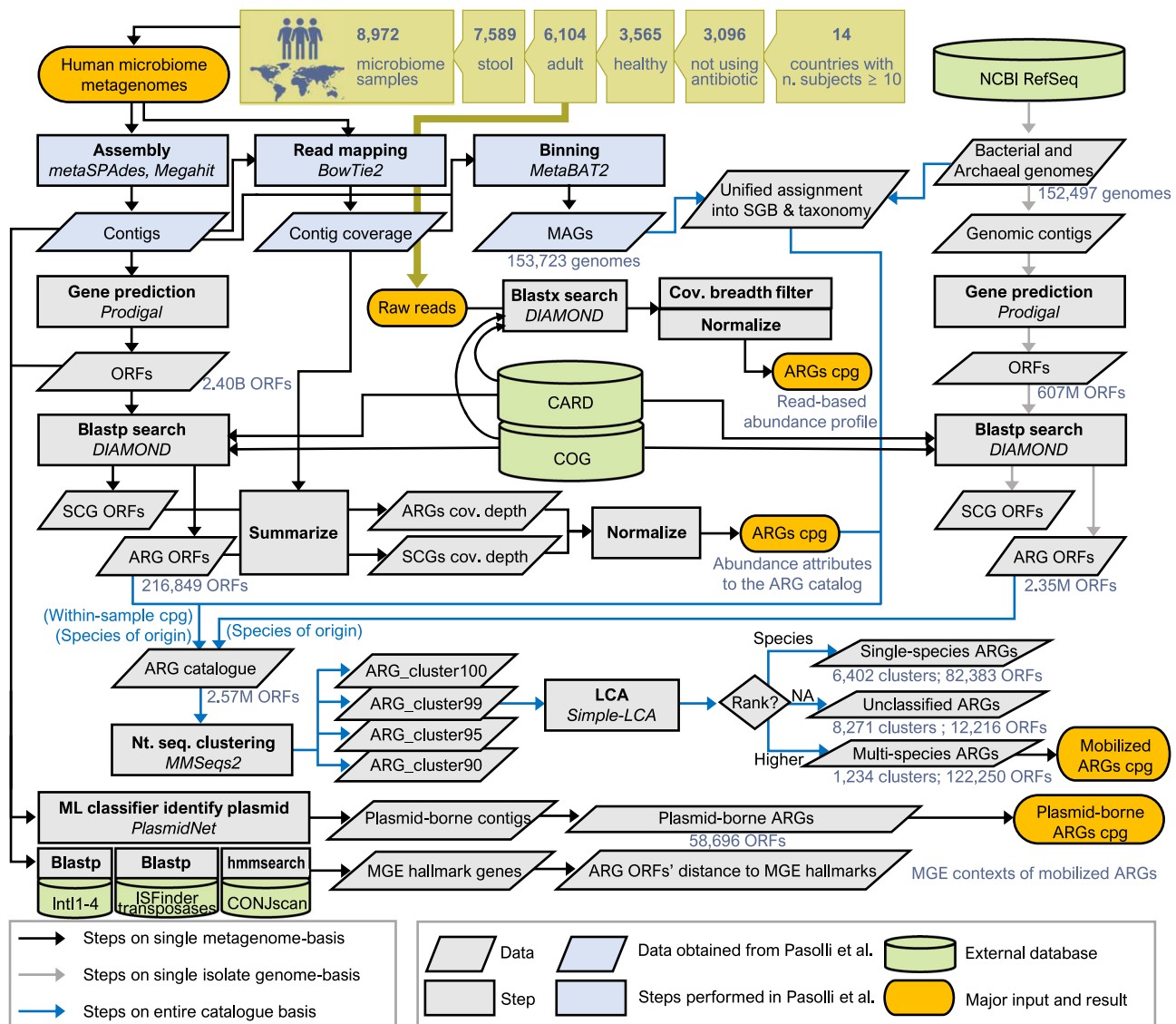

**Fig. 1 | A schematic overview of the bioinformatics pipeline employed in the study.** This summarises the overall strategy used to profile ARGs in the microbiome samples and compare to the genome collection, for details see the Methods. The number of samples used for each country is given in Table S1 and the number of ARG clusters created in Table S2.

4.7, these differences were significant (Kruskal-Wallis, $\chi^2(3) = 306.13$, $p = 4.7e-66$). However, to account for variable sample size we also scaled by total bases sequenced (see Methods), then airways had the highest richness of 289.9 ARG_cluster99s per 100Gbp of sequence (Kruskal-Wallis, $\chi^2(3) = 228.36$, $p = 3.1e-49$), compared to stool with 263.3, skin=206.4 and oral cavity=189.0 (see Figure S2a, b for ARG_cluster99 rarefaction curves). The two body sites best represented in our study after the gut were the oral cavity and skin. Oral and skin datasets only shared 16.6% and 52.0% of their ARG_cluster99s with the gut catalogue, implying that each body site has a distinct resistome. Using a lower resolution ARG clustering, ARG_cluster90, the overlap increases to 77.8% and 74.1% of their catalogues respectively, but there are still ARGs not present in the gut samples.

## ARG diversity and abundance of the healthy gut microbiome varies across countries

The gut microbiome is in general the best studied component of the human microbiome and in this data set only the stool samples spanned a wide range of different countries. We have 23 countries with greater than ten samples for stool as compared to four for skin, three for the oral cavity and just one for the other sites. We therefore restricted further analysis to stool samples, excluding infants and children because of the known instability of the gut microbiome in early life[33], and filtering low quality samples (see Methods). This gave 6104 samples spanning twenty different countries. In Fig. S2c, we give rarefaction curves for the ARG gene diversity as a function of total amount of sequencing summed across samples at different clustering cut-offs. This confirms that even though the 6104 samples totalled nearly 30 Tbp of sequenced reads, there is no sign of ARG diversity reaching an asymptote even for the 90% identity clusters. Therefore, even though the ARG catalogue we present here is the most comprehensive to date, it represents only a fraction of the true diversity of ARG sequences in the gut microbiome.

These samples include both healthy individuals and those who have at least one diagnosed disease, more precisely: 3565 from healthy controls, 1658 from subjects labelled with a specific disease, 131 from miscellaneous cases including hunter-gatherers and noncontemporary samples, 750 from subjects without health-disease information (for detailed sample information see Supplementary Data 1 and Table S1). As we would expect disease to impact the overall microbiome state and possibly associate with increased antibiotic usage, which in turn may reshape the ARG profiles, we restrict the following analysis to the 3565 healthy control samples. We then kept only the healthy controls that could be unambiguously determined to be not taking antibiotics at the time of sampling, to give a data set of 3096 samples. It is important to note that because this is a meta-analysis there is no single definition of healthy control which might vary from one study to another.

The fact that ARG diversity is not saturating, implies that for ARG richness comparisons between countries we must correct for the varying sample number and sizes. Therefore we developed and tested a procedure that generated random subsets of samples from each country comprising $100 \pm 10$ Gbp of sequence (see Methods and Table S3). Following this subsampling procedure we observed significant differences in diversity across countries at all cut-offs (Kruskal–Wallis, $\chi^2(10) = 877.33$, $p = 7.7e-183$). The country with the highest median subsampled ARG_cluster95 diversity per 100 Gbp of sequence was China (222) with approximately four times the diversity of the lowest median diversity, observed in the USA (55), Fig. 2d (Wilcoxon rank sum, $W = 9801$, Benjamini–Hochberg adjusted B.-H. $p = 2.6e-33$).

The above analysis was based on annotating metagenome assemblies. This is necessary to determine the number of novel ARGs in a sample. However, for simply profiling the abundance of ARGs we found read-based mapping directly to the CARD to be more sensitive

than assembling first (see Fig. S3). This is probably because a minimum coverage depth for a gene will be necessary for assembly to be possible. Following read-mapping abundance levels were expressed as 'copies per genome (cpg)', calculated for each ARG in each sample as the coverage depth normalized with respect to a panel of prokaryotic single-copy core genes (SCGs - see Methods). The summed abundance of all ARGs in the healthy subjects without current antibiotic usage ($n = 2,740$ profiled by the read-based approach) ranged from 0.751 to 18.4 (5th–95th percentile) with a median of 2.71 cpg. When the summed ARG abundances in the samples were grouped by country and the medians in the 14 countries with at least 10 samples from healthy individuals were compared to each other, we found significant variation between countries (Fig. 2a Kruskal–Wallis, $\chi^2(13) = 920.16$, $p = 2.4e-188$, B.-H. adjusted $p < 0.05$ for 73 out of 91 country pairs tested). There was a five-fold variation in median resistance levels between the lowest in the Netherlands with 1.08 cpg to Spain with 5.56 cpg (Wilcoxon rank sum, $W = 1130$, B.-H. $p = 1.6e-65$).

## Gut ARG abundance in healthy individuals correlates with antibiotic consumption

We then compared two population-level measures of the potential impact of antibiotics on the gut microbiome of a country, the median rarefied ARG_cluster95 richness and the median total prevalence of ARGs (cpg), with the antibiotic consumption rate (Fig. 2). We relied on two resources for the data on national antibiotic consumption rates: ResistanceMap operated by the CDDEP[34] and the WHO report[35]. We used the total defined daily dose (DDD) per 1000 (capita) per year – shortened to DDD per 1000 – summed across all antibiotic classes surveyed from each data resource, to quantify the overall intensity of antibiotic usage in each country. Of the 20 countries for which we could quantify the resistome (i.e. at least 10 high-coverage metagenomes available), CDDEP data covered 15 countries and WHO data covered 14 countries, 12 countries were covered in both, three countries were not included in either. The two sources of data did not completely agree, although there was a strong correlation in those 12 countries where they overlapped (Fig. S4 Pearson's correlation, $r = 0.77$, $p = 0.0036$).

We found significant correlations between a country's ARG abundance, as measured by the median of the per-sample ARG copies per genome, and per capita antibiotic usage rates (Fig. 2b, c; Pearson's correlation $r = 0.89$, $p = 2.3e-5$ for CDDEP consumption rates excluding China, and $r = 0.65$, $p = 0.040$ for WHO usage rates). In the CDDEP comparisons China is a notable outlier and we discuss possible reasons for this below. WHO data was not available for China. We also observed a strong positive correlation between the rarefied ARG richness and the antibiotic usage rates for the WHO statistics for ARG_cluster95 ($r = 0.86$, $p = 0.0063$) although for the CDDEP it was only marginally significant (see Fig. 2e-f: $r = 0.57$, $p = 0.11$). We used Pearson's correlation for these comparisons as both the total ARG abundance and the rarefied diversity appeared Normal under a Shapiro-Wilk test (see Table S4). When we separated the abundance and diversity of ARGs into different classes of antibiotics and correlated those individually with the consumption rate of the corresponding class, in most cases we no longer found correlations, with the notable exception of the Beta-lactams where for abundance we did observe a significant correlation for the WHO data ($r = 0.73$ and $p = 0.021$ – see Table S5).

## Taxonomic assignment of ARG clusters and identification of mobile ARGs using a human microbiome MAG collection and reference isolate genomes

Bacterial genomes contain intra- and extra-chromosomal MGEs which facilitate rapid horizontal gene transfer - HGT[4,36]. This enables them to evolve rapidly under selective pressures. This horizontal gene transfer can occur both within species and between more distantly related

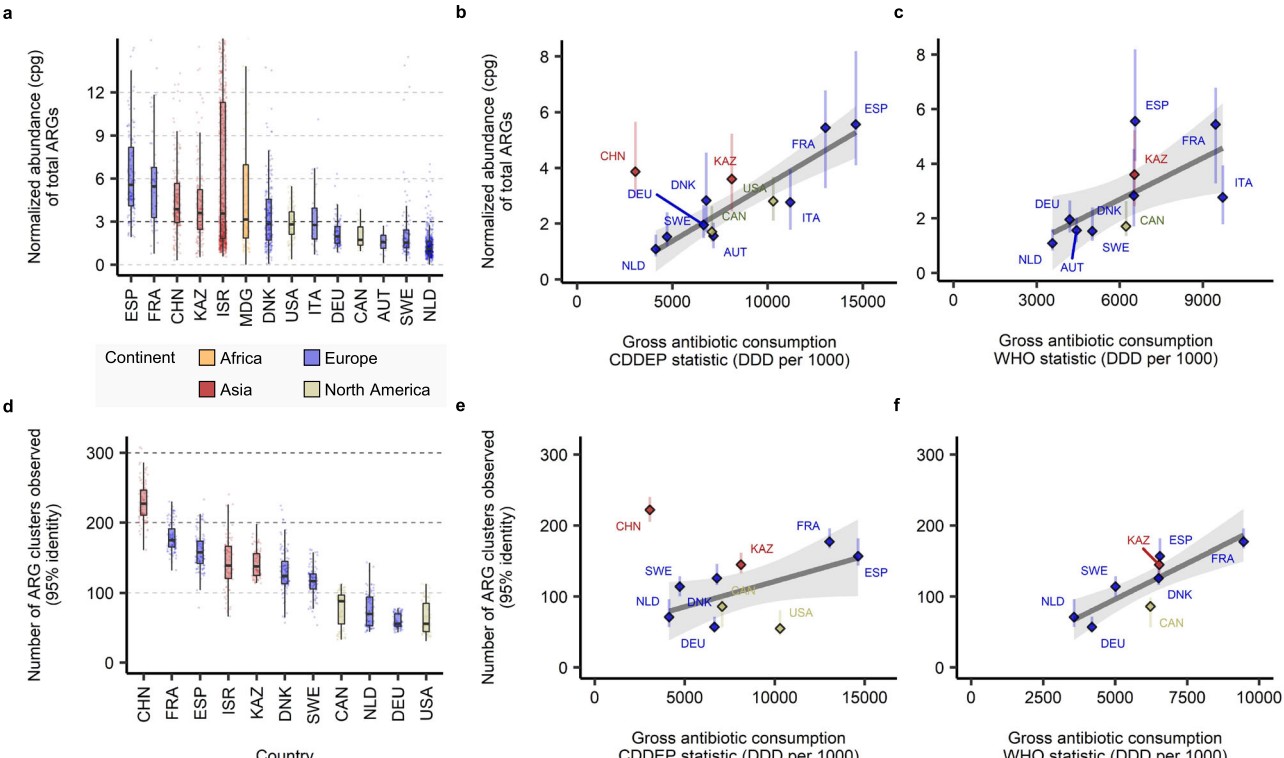

**Fig. 2 | Correlation between median diversity and abundance of ARGs in healthy adult gut metagenomes in a country and antibiotic consumption rates. a** Median total gut ARG abundance (cpg) in healthy antibiotic free adults for countries with >10 samples. **b, c** Correlations between median total ARG abundance and the per capita rate of antibiotic consumption (DDD per 1000), using CDDEP consumption statistics in (**b**) and WHO in (**c**). Pearson's correlation tests gave $r = 0.89$, $p = 0.00023$ for CDDEP (**b**) and $r = 0.65$, $p = 0.040$ for WHO (**c**), excluding China from the CDDEP correlations. **d** Median rarefied (100 ± 10 Gbp subsamples) ARG_cluster95 richness across countries. **e, f** Correlations between ARG_cluster95 richness per 100 Gbp and the per capita rate of antibiotic consumption for CDDEP (**e**) and WHO (**f**) estimates. Pearson's correlation tests $r = 0.57$, $p = 0.11$ for CDDEP (**e**) and $r = 0.85$, $p = 0.0073$ for WHO (**f**), excluding China from the CDDEP correlations. Vertical lines range from the 25th percentile to 75th percentile (**b, c, e, f**). Country-level ARG statistics are given in Table S3. Correlation

tests in Table S4. Novelty of the ARG clusters with respect to RefSeq is visualized in Fig. S1. Rarefaction curves of ARG cluster richness in human microbiomes are shown in Fig. S2. . Linear trend line was determined by a generalized linear model using ggplot2 R package, the shaded area represents 95% confidence interval. Country name abbreviations and metagenome sample number: AUT ($n = 16$) - Austria, CAN ($n = 35$) - Canada, CHN ($n = 209$) - China, DEU ($n = 103$) - Germany, DNK ($n = 230$) - Denmark, ESP ($n = 139$) - Spain, FRA ($n = 62$)- France, ISR ($n = 937$) - Israel, ITA ($n = 33$) - Italy, KAZ ($n = 168$) - Kazakhstan, MDG ($n = 112$) - Madagascar, NLD ($n = 468$) - Netherlands, SWE ($n = 109$) - Sweden, USA ($n = 115$) - United States. The number of rarefactions performed to derive each box plot and range bar shown in **d–f**: $n = 99$. In the box plots shown in **a** and **d**, the box spans from 25th to 75th percentiles, the line inside the box is the median, and the whisker spans from the minimum to the maximum values.

organisms even across phyla[20,37,38]. We employed two independent approaches to determine which of the ARGs in our resistome catalogue have been recently mobilized. Firstly, we searched for highly similar (99% nucleotide identity) ARG clusters, which were found in two or more distinct taxa. With this approach we exploited the fact that our ARGs derive from the same metagenome data used by Pasolli et al. (2019) to generate a collection of 154,723 human microbiome MAGs[26]. These MAGs were assigned along with the RefSeq prokaryotic genomes to species-level genome bins, SGBs, at 95% average nucleotide identity (ANI) with a complete taxonomy to superkingdom ranks. This enabled us to assign species labels to any ORF that was derived from MAGs or RefSeq genomes and perform lowest common ancestor assignments (LCA) across the ORFs present in an ARG cluster. We then labelled any ARG_cluster99 that was found in multiple SGBs as 'multi-species' and those found in only one SGB as 'single-species', ARG_cluster99s without any SGB-assigned ORFs were 'LCA-unassigned'. This strategy of searching for highly similar (99% nucleotide identical) sequences across species to detect recent horizontal gene transfer is equivalent to that used in[39].

We were able to assign taxa to 54.5% of the ARG_cluster99s and 10% of the ARG_cluster99s were assigned to multiple species across the whole data set. Despite this, the putatively mobile multi-species ARGs constituted the majority of each individual's gut resistome: 87% of

within-sample ARG richness and 96% of the total ARG abundance (cpg) per individual. Note ARG cpg here is calculated using the assembly approach since our definition of mobility requires assembly. Multi-species ARGs were biased towards particular antibiotic classes and mechanisms, the highest ratio of recently mobilised clusters was found for the sulfonamides and diaminopyrimidine, and they were more prevalent amongst target replacement and inactivation resistance mechanisms (see Table S6 and Fig. S5c). The multi-species ARGs were geographically more widespread, *i.e.* dispersed across more countries, than the single-species ARGs (average number of countries 5.9 vs. 3.0, Wilcoxon rank sum, $W = 3478595$, $p = 1.21e − 74$).

To complement the above identification of mobile genes through their presence in multiple taxa we also directly searched for sequences characteristic of MGEs and determined which ARGs are on or in close proximity to them. For plasmids we did this using a machine learning approach that uses features such as ORF length and dipeptide frequencies (see ref.[30] and Methods). To identify ARGs possibly mobilised by other MGEs, we searched for the hallmark genes of insertion sequence (IS) elements, conjugative elements (e.g., ICEs), and class 1 integrons among the genomic and metagenomic ORFs and subsequently classified each ARG-annotated ORF using MGE-specific thresholds of distance on genomic contigs. Applying this strategy to the gut metagenomes, we found that overall 36.2% of the adult gut

resistome ORFs were associated with at least one type of MGE (27.1% on plasmids; 4.4% proximate to IS transposases; 10.2% proximate to conjugative systems; 0.3% proximate to integron integrase). The proportion of MGE-associated ARGs in the RefSeq-derived catalogue was 39.2%. There was a good correspondence between ARG localisation on a plasmid or being positioned near a MGE hallmark gene and the definition of recently mobilised ARG clusters described above (see Fig. S5a and Fig. S6). Among the ARG-annotated ORFs in the adult gut metagenomes, 89.1% of the plasmid-borne ARGs were found within multi-species clusters, compared to the 49.0% of non-plasmid-borne ARGs. Similarly, 80.9% of ICE-associated ARGs were multi-species, 82.1% of the ARGs located ≤100 Kbp from IS transposases, and 99.4% of the ARGs located ≤10 Kbp from integron integrases. This enrichment of multi-species ARG clusters in MGE-associated ARGs is logical as these ARGs are expected to have a higher chance to cross species boundaries into new hosts. Conversely, 53.0% of multi-species cluster ORFs were associated with at least one MGE type, whilst of the single species cluster ORFs only 12.5% were. ARGs involved in rifamycin, nucleoside and sulfonamide resistance were most frequently located on plasmids (≥90% of ORFs) while fluoroquinolone, fosfomycin, peptide, and multi-drug resistance genes were infrequently located on plasmids (<10%) − See Fig. S5b.

We then separated both the total abundance and the richness of ARG clusters in each country across these gene mobility categories. The non-mobile species-specific gene abundances in a country did not have a significant association with per capita antibiotic consumption but both the plasmid-borne ARGs and the multi-species mobile clusters showed similar and in some cases stronger correlations of abundance with consumption rates than the total resistance (see Table S4 and Fig. S7). The same was not true for the ARG diversity, as measured by rarefied ARG_Cluster99 richness, here we do see correlations across all gene types, we will return to this observation in the Discussion.

### ARG clusters are preferentially shared between phylogenetically similar species that co-occur across gut microbiome samples

For each candidate species SGB we determined the total number of unique ARG_cluster99s observed across all MAGs and reference genomes assigned to that species. We then considered every pair of species and counted the number of clusters that were shared between them. Using a negative binomial regression we found a highly significant negative association (coefficient= − 0.51 ± 0.003, $p < 2e − 16$) between the rate of ARG cluster sharing and the phylogenetic distance but a positive association (coefficient=2.43 ± 0.02, $p < 2e − 16$) with the number of samples they co-occur in (see Methods). These coefficients correspond to an eleven fold increase in the rate of gene sharing as species go from never co-occurring to being found in every sample together and a 40% decrease in gene sharing for every unit of phylogenetic distance (maximum in the data set 4.55). This matches with earlier studies on HGT[37].

### Human gut resistome profiles exhibit two distinct 'resistotypes'

We characterised the resistome composition for each adult gut microbiome as a profile giving the total normalised (cpg) abundance of each ARG family in a curated version of the CARD database using a read-based approach (see Methods). We observed 422 out of 752 total ARG families at least once in the 5372 adult gut metagenomes for which the raw reads were processed. These 422 dimensional profiles were then mapped onto two dimensions using NMDS (see Methods) and Bray-Curtis distances. Two distinct clusters were observed separated by the first NMDS axis (Fig. 3a). This apparent binary split was confirmed quantitatively by performing partitioning around medoid (PAM) clustering using Bray-Curtis dissimilarities calculated on all 422 ARG family abundances (see Methods). Increasing cluster number ($k$) from 2 to 20, we found that the average silhouette score, a measure of how well separated the clusters are, was maximized for two clusters

(Fig. 3b). The observation of two clusters was robust to the choice of distance measure (Manhattan and Euclidean also tested) in PAM clustering (Fig. S8a, b) and clustering method, k-means with the elbow method also predicted two clusters (Fig. S8c). Finally, UMAP an alternative projection method also generated two recognizable clusters of resistome profiles (Fig. S8d).

We will refer to the two clusters derived from PAM clustering with Bray−Curtis dissimilarities as 'resistotypes'. In order to give a quantitative measure of how likely a profile is to derive from each cluster, we define a 'resistotype scale index' (RSI) as the difference of the Bray−Curtis dissimilarities to the medoids of the resistotype PAM clusters. The RSI displayed a clearly bi-modal distribution that is consistent with the PAM assignment (Fig. 3c). There is a higher frequency of one resistotype (56.5% vs. 43.5%; $n = 5372$) and more so within the healthy subjects (67.1% vs. 32.9%; $n = 3113$). The individuals in the less common resistotype displayed higher overall ARG abundance (Fig. 3d; median cpg 2.04 in major, 3.72 in minor, fold difference=1.82; Wilcoxon rank sum, W=2276161, $p$=1.49e-112) and greater ARG cluster richness (Fig. S9a, b) compared to individuals in the high frequency resistotype. We calculated the total ARG abundance (cpgs) in each antibiotic class for the samples and compared these between the two resistotypes. We found that the low frequency resistotype was more than 10-fold enriched for multiple antibiotic classes and the five with the highest fold difference were fluoroquinolones, fosfomycins, aminoglycosides, and peptide antibiotics, as well as multi-drug resistance genes (Fig. 3d and Table S7). In contrast, ARGs in these classes were almost absent in the major resistotype. Based on the initials of these antibiotic classes, we will refer to the low frequency resistotype as the 'FAMP' resistotype, and the high frequency resistotype as the 'background' resistotype.

### Resistotypes are independent of enterotypes but are associated with particular species

We then determined the degree to which the resistotypes can be explained by the species composition of the gut microbiome or alternatively as a function of mobile multi-species genes. We calculated the species abundance profile for each sample by identifying single-copy core genes (SCGs) associated with SGBs (see Methods). The number of species detected in the adult gut metagenomes was distributed around a median of 329.5 species (inter-quartile 268−395) for healthy adult subjects (Fig. S9c). Overall, there was an association between species profile and resistance profile as ARG family abundances (Mantel test $r = 0.27$, $p < 0.001$).

We next determined which species were associated with the two resistotypes (see Fig. 3e and Table S7). SGBs that are more abundant in FAMP resistotype individuals, included opportunistic pathogen species such as *Escherichia coli* (log10-fold mean difference - $Log10FMD = 4.4$, Wilcoxon rank sum, $W = 858917$, $p = 0$). and *Proteus mirabilis* ($Log10FMD = 4.2$, $W = 891027$, $p = 0$), whereas beneficial gut anaerobes (e.g., butyrate producers or complex carbohydrates degraders) such as *Coprococcus eutactus* ($Log10FMD = 0.88$, $W = 4132374$, $p = 1.1e − 57$) and *Eubacterium siraeum* ($Log10FMD = 0.86$, $W = 3858630$, $p = 2.8e − 28$) were more abundant in the background resistotype samples. In the following section, we determine systematically, whether the species associated with each resistotype are enriched for commensal or pathogenic bacteria.

The species configuration of microbiomes has previously been proposed to cluster into distinct 'enterotypes'[40,41]. We searched, therefore, for correlations between these enterotypes and the resistotypes identified here. In confirmation with the enterotype studies we did find three clusters of species configurations, corresponding to the clusters conventionally characterised by the dominance of *Prevotella, Bacteroides*, and *Lachnospiraceae* (or *Firmicutes*), respectively (see Fig. S8f, g and Table S8). There was an association between enterotype and resistotype (Chi-sq. test, $\chi^2(2) = 65.185$, $p = 7.0e − 15$), but it was

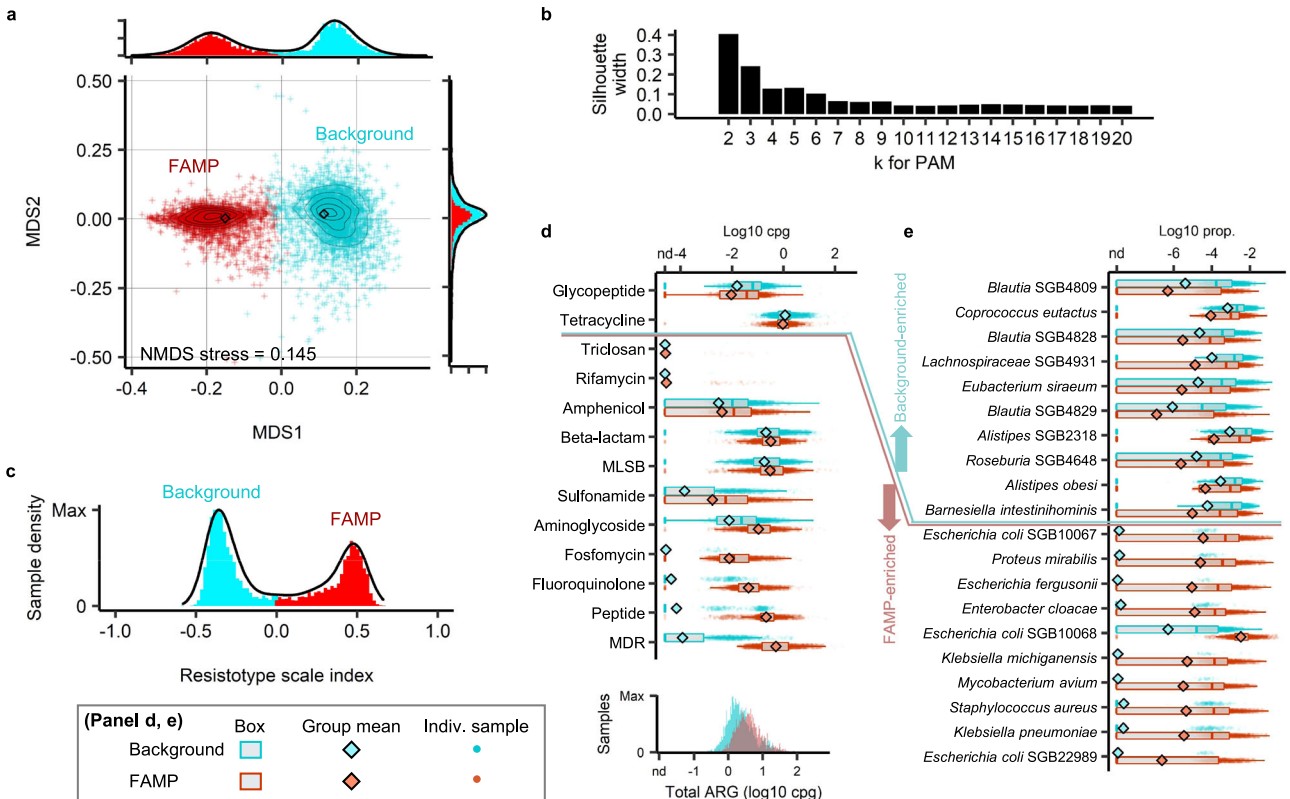

**Fig. 3 | Two distinct clusters apparent in the global landscape of adult gut resistome profiles. a** NMDS projection of Bray–Curtis dissimilarities among the log-transformed ARG family profiles (cpg) in adult gut metagenomes (contours estimate sample densities). Samples were colored by cluster assignment (PAM, Bray-Curtis clustering, k=2). **b** Average silhouette width of PAM Bray-Curtis clusters as a function of cluster number k. **c** Sample density projecting points onto the line joining cluster medoids using Bray–Curtis dissimilarities. **d** Box plots of the summed abundance of ARGs in each antibiotic class, separated by resistotype (`background' or `FAMP') with the distribution of total ARGs (cpg) by resistotype shown at the bottom. **e** Relative abundances of the ten species with highest mean fold difference between resistotypes and two-sided Mann–Whitney test,

Benjamini–Hochberg-adjusted $p < 0.05$. Statistics on the differential abundance of ARG classes and the species between the two resistotypes are provided in Table S7. Results from clustering the ARG profiles with alternative methods and clustering of species compositions are given in Fig. S8. Comparison of species diversity and ARG diversity between the background and the FAMP resistotypes is provided in Fig. S5. Naming of the FAMP resistotype reflects the five antibiotic resistance classes enriched with the largest fold differences: F, fluoroquinolone and fosfomycin; A, aminoglycoside; M, multi-drug; P, peptide. Number of metagenome samples $n = 3034$ for the background resistotype, $n = 2338$ for the FAMP resistotype. In the box plots, the box spans from 25th to 75th percentiles, the line inside the box is the median, and the whisker spans from the minimum to the maximum values.

quite weak (Cramer's $V = 0.11$), driven by slight differences in resistotype proportions in each enterotype, e.g. the *Lachnospiraceae* enterotype being less likely to have the FAMP resistotype (Fig. S8h).

This suggests that each resistotype is not associated with a particular species profile such as an enterotype but rather is driven by a subset of the microbiome. This was confirmed by the fact that just 1% of the variance in species profile depends on resistotype (Bray–Curtis perm. ANOVA, $R^2 = 0.011$, $p = 0.001$). To resolve whether the FAMP resistotype might instead reflect a low diversity dysbiotic state we compared the species richness of the individuals assigned to the background and FAMP resistotypes but found the opposite trend, a slightly higher species diversity for the FAMP samples (359 vs. 336 median SGBs; Wilcoxon rank sum, $W = 2853218$, $p = 6.3e-15$), particularly for the *Proteobacteria* (Fig. S9d).

### Pathogenic and non-resident species are associated with the FAMP resistotype

The gut microbiome comprises a complex community of both potentially pathogenic and harmless or beneficial resident commensal organisms. It is the pathogenic species, more specifically certain strains encountered in clinical infections, that will be the target of antibiotic treatment. We might expect ARGs associated with pathogens to drive both the response to country-wide antibiotic consumption and the distinct resistotypes we observed above. To

determine the type of organism an ARG cluster is associated with, we first classified SGBs as either pathogenic or non-pathogenic, depending on whether any strain has been reported to cause infection at any site in the human body including the gut itself[42]. This species level definition of potential pathogenicity will include opportunistic pathogens, where only particular strains in particular circumstances are pathogenic, but since they will then be the target of antibiotic treatment and developed resistance may spread relatively easily to other strains of that species it seemed a pragmatic definition. On this basis, 237 of the 4686 SGBs that occurred in our adult stool metagenome profiles (roughly 5%) were classified as pathogens. In Fig. S10a we show the distribution of prevalences (percentage of samples they occur in) for pathogenic and non-pathogenic SGBs across the 3,096 gut microbiome samples from individuals that were healthy and not taking antibiotics at the time of sampling (see Methods). For both pathogens and non-pathogens a broad distribution of prevalences is observed with some opportunistic pathogens e.g. *Bacteroides ovatus* and *B. thetaiotaomicron* being found in the vast majority of samples. We therefore further classified SGBs as either resident (present in ≥10% of samples) or an infrequent colonizer (<10% of samples). There is a weak negative association between pathogenicity status and residency status, with 11.4% of pathogens being classed as resident versus 16.1% of non-pathogenic species (Fisher's exact test, $p = 0.055$, see Table S9).

We next defined an SGB as associated with one of the two resistotypes if the mean abundance was at least five-fold higher in one than the other and this difference was significant (Mann–Whitney test, adjusted $p < 0.05$). We found that the SGBs associated with the FAMP resistotype were enriched for pathogen SGBs (41.2% were pathogens) compared to the background resistotype-associated SGBs (0% pathogens) or the SGBs not associated with either resistotype (4.8% pathogens), and overall, there was a highly significant association between pathogen status and resistotype association (see Table S9, Fisher's exact test, $p = 5.2e - 10$). Similarly, the FAMP associated SGBs were biased toward species with lower prevalences, i.e., infrequent colonizers (Fig. S10).

We investigate the distribution of ARG families across SGBs associated with the two resistotypes further in Fig. 4. ARG families are ordered by their association with the FAMP resistotype. This confirms that the majority of FAMP associated families are found in *Proteobacteria* particularly *Enterobacteriaceae*, that they are mostly core to the species and that the most strongly associated are actually neither mobilised across multiple-species or plasmid-borne. The background resistotype associated SGBs are distributed throughout the other major phyla.

## Country-level response to antibiotic consumption but not resistotype is driven by ARGs that are shared between pathogens and resident commensals

We can define the single-species ARG_cluster99s in terms of the type of species - pathogen, non-pathogen or neither - that they are found in, multi-species clusters can additionally derive from both. This reveals that far more of the ARG abundance in the FAMP resistotype derives from pathogen associated clusters (33.0%) either single or multi-species rather than the background resistotype (0.0%). In contrast, for the major resistotype the most important class by coverage (89.3%) are multi-species clusters that span both pathogens and non-pathogens (Fig. S10c). We can refine this further by considering the taxonomy of the associated SGBs and again separating by cluster type and resistotype (see Fig. S10d). From this we see that the majority of the ARG coverage in the FAMP resistotype derives from *Proteobacteria*, which is not the case for the background resistotype, where clusters are mostly shared across phyla, as was seen at the ARG family level in Fig. 4.

We then used the same ARG_cluster99 cluster assignments to determine the type of ARG that is driving the correlations between ARG abundance and antibiotic consumption rates presented above. To do this we summed the abundance of all clusters deriving from the following four ARG categories: pathogen-associated, pathogen and resident non-pathogen (*i.e.* commensals), just commensals and unassociated. We then repeated the correlation analysis between the median ARG abundance in each country with antibiotic consumption rates for each category of ARG. The results are given in Fig. 5, significant correlations are only found for the ARGs that are shared between pathogens and commensals (Fig. 5c WHO - r = 0.81, $p = 4.2e - 3$ and Fig. 5g CDDEP excluding China - r = 0.81, $p = 2.6e - 3$).

## The FAMP resistotype is associated with antibiotic exposure and some diseases, particularly enteric infections

The frequency of the FAMP resistotype is quite strongly (Cramer's $V = 0.248$) associated with disease status (Chi-Sq. test, $\chi^2(11) = 285.23$, $p = 1.1e - 54$ - Fig. S11a). There appears to be a gradient from healthy individuals with 32.9% FAMP (22.8% among the samples from the 1980s) through colorectal cancer (51.9%) and metabolic disorders (58.2%) to enteric infections with Shiga-toxin-producing *Escherichia coli* at 79.4% and cholera with 83.3%. In healthy individuals, recent antibiotic exposure within the last three months, is associated with a higher frequency of the FAMP resistotype (Fig. S11b, (Chi-Sq. test, $\chi^2(1) = 5.5293$, $p = 0.019$), and this association is stronger if we include non-healthy subjects too (Chi-Sq. test, $\chi^2(1) = 12.17$, $p = 0.00049$),

probably because of the increase in antibiotic positive sample number from $n = 38$ to $n = 244$. It is important to note though that the FAMP resistotype is still present in 28.6% of healthy subjects not currently exposed to antibiotics.

There is no difference between sexes (Chi-Sq. test, $\chi^2(1) = 1.8586$, $p = 0.173$). There was, however, a significant but weak positive association between age in years and the probability of deriving from the FAMP resistotype (logistic regression of resistotype against age for healthy samples: $n = 1,267$, coeff. = 0.0056, $p = 1.3e - 11$). There were also strong (Cramer's V = 0.449) significant differences in resistotype frequencies across countries for these samples (Chi-Sq. test, $\chi^2(13) = 552.28$, $p = 1.1e - 109$) but the frequency of the FAMP resistotype was not found to be correlated with the antibiotic consumption rate (Pearson's correlation for WHO statistics, $n = 10$, $r = 0.37$ and $p = 0.29$; CDDEP statistics, $n = 12$, $r = 0.097$ and $p = 0.76$).

## Consumption of antibiotics causes a short-term shift to the FAMP resistotype

We used existing time series data from healthy individuals in a controlled antibiotic trial[11] to explore the short-term dynamics of resistotype classifications. In this study, twelve men received a cocktail of three last-resort antibiotics orally, meropenem, gentamicin and vancomycin, for four days and then their gut microbiome was tracked for six months. We calculated the normalized abundance profiles of ARGs from the metagenome shotgun reads generated in this study, as described above, and then used our 'resistotype scale index' (RSI) to determine how similar each sample's resistome profile was to our resistotypes. We observe a clear shift to the FAMP resistotype at eight days after the end of the antibiotic treatment (see Fig. 6a). This effect persisted slightly at 42 days, although by 180 days most individuals had entirely returned to the background resistotype. The observed transitions to the FAMP resistotype following antibiotic exposure was accompanied by an increase in total ARG abundance (Fig. 6b). This temporary increase in ARG abundance was attributable to single-species rather than multi-species ARGs (Fig. 6c) deriving from *Proteobacteria* (Fig. 6d).

## ARGs driving country-level response are embedded within a closely connected component enriched for resident non-pathogen species

We constructed a weighted bipartite graph of ARG clusters and SGBs by linking an ARG to a SGB if any ORFs assigned to that ARG_cluster99 were found in a high-quality MAG assigned to that SGB (see Methods). Links supported by a single ORF were removed and each remaining link was weighted by the number of ORFs supporting the link. In Fig. 7a we visualise this network and colour nodes by their status as pathogens (red) and non-pathogens (blue). Bipartite module detection using Beckett's method[43] revealed 10 subnetworks: a single central module containing a large number of species, mostly non-pathogenic (91.1%) and gut microbiome residents (81.6%), which are strongly interconnected via multi-species ARGs, and nine peripheral modules (Fig. 7a, b).

To simplify interpretation we will compare this central module to all the peripheral modules grouped together. The majority of SGBs (158 out of 229, i.e. 70.0%) were assigned to the central module but the majority of ARG_cluster99s, were found in the peripheral modules (1284 out of 1447, i.e. 88.7%). The SGBs in the central module had a higher proportion of non-pathogens compared to the peripheral modules (91.1% vs. 76.1%, Chi-sq. test, $\chi^2(1) = 8.28$, $p = 0.004$) and also more resident gut species (81.6% vs. 62.0%, Chi-sq. test, $\chi^2(1) = 9.23$, $p = 0.002$). In addition, more SGBs associated with the FAMP resistotype were found in the peripheral modules (11.3%) compared to the central component (3.16%; $p = 0.032$; Fig. 7b) whilst the SGBs associated with the background resistotype were roughly evenly split between the two (4.43% vs.

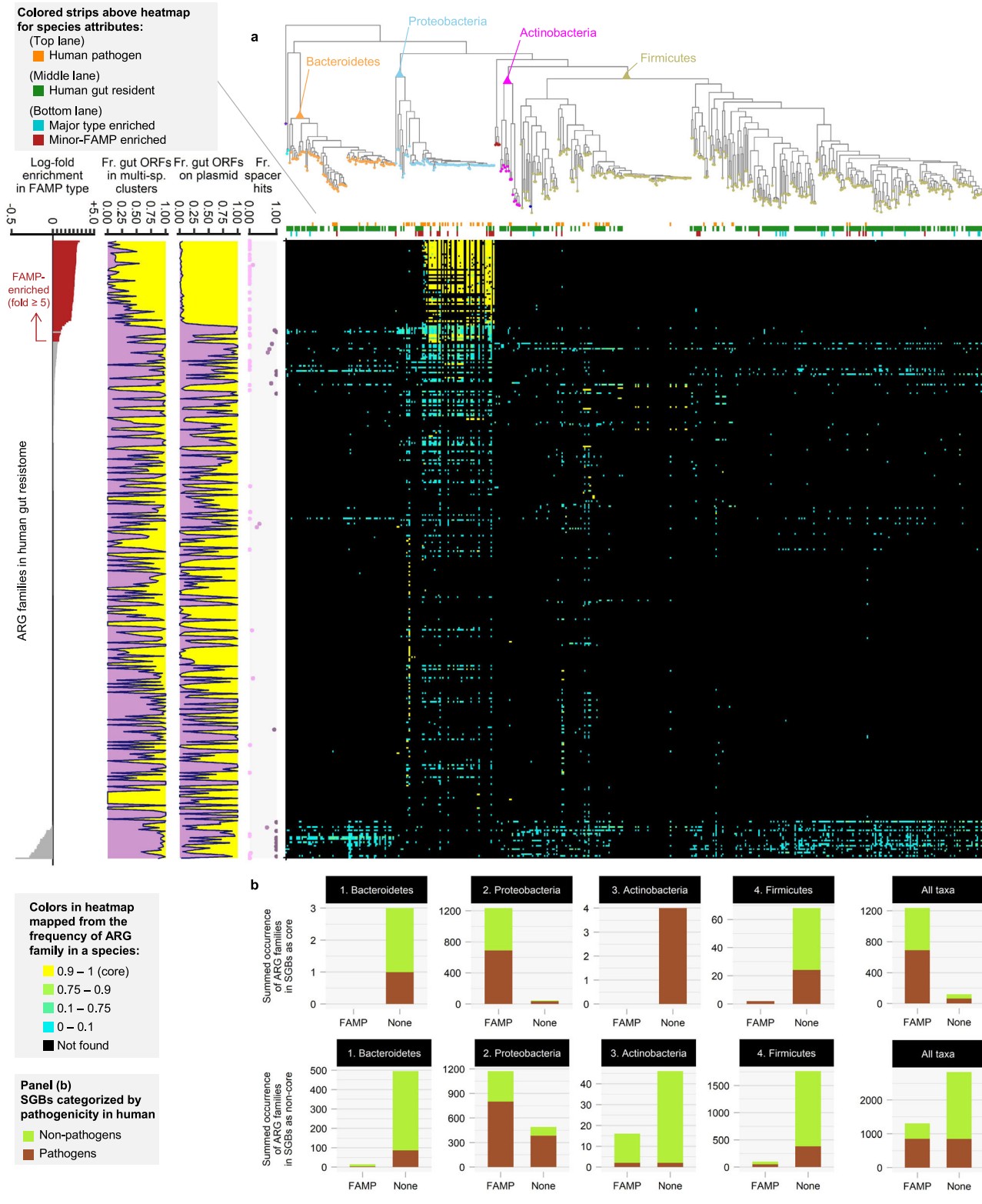

2.82% of the SGBs in the central and the peripherals, respectively; $p = 0.83$).

The ARG clusters also differ between the central and peripheral modules, there is a much higher preponderance of multi-species clusters in the central module (57.1% of ARG_cluster99 vs. 17.4%) with single-species clusters dominating in the periphery (82.6% vs 42.9%, Chi-sq. test, $\chi^2(3) = 131.15$, $p = 2.3e-30$ and Fig. 7c). The same is true of plasmid associated clusters which are more often found in the central

module. We see the same patterns for ARGs as SGBs with a higher percentage of pathogen and non-resident associated ARGs in the peripheral modules. The peripheral modules are strongly enriched for FAMP associated ARG families, with 85.7% of the ARG_cluster99s in the periphery deriving from these families compared to 21.5% for the central module (Chi-sq. test, $\chi^2(1) = 351.39$, $p = 2.11e-78$; Fig. 7c). In contrast, the central module has a higher percentage of clusters that correlate with the country-level per-capita antibiotic consumption

**Fig. 4 | Phylogenetic distribution of the ARG families in the two resistotypes.** Main panel (**a**): heat map gives presence of each ARG family across the phylogenetic tree of the SGBs, color reflects the fraction of genomes in the SGB that contain the ARG family. The 363 ARG families detected in adult stool metagenomes based on our assembled catalogue are sorted (y-axis) by fold-difference between the mean abundances in the background and FAMP resistotypes. The phylogenetic tree includes the 522 SGBs which are most abundant in the gut microbiome or are associated with resistotypes (see Methods). Subpanels left to right: bar plot of the fold-difference in abundance between the resistotypes; fraction of ORFs in multi-species ARG_cluster99s in each ARG family; fraction of plasmid-borne ORFs; fraction of ORFs containing at least one high-identity alignment of CRISPR spacers

collected from gut metagenomes (minimum 90% identity over 90% of the spacer length) in each ARG family. Panel (**b**): For each major phylum and for all taxa combined, we give the total occurrence of ARG families in the core resistomes of the SGBs (top row) and the accessory resistomes (bottom row). FAMP-associated and non-associated ARG families were compared, and the occurrence in pathogenic and non-pathogenic species were coloured differently. The maximum-likelihood phylogenetic tree was reconstructed using one representative genome for each of the selected SGBs. Concatenated nucleotide sequence alignments of 40 single-copy COGs were used as the input. The overall contribution of ARG subsets based on pathogenicity and residence to the resistomes of background and FAMP reistotypes is summarized in Fig. S10.

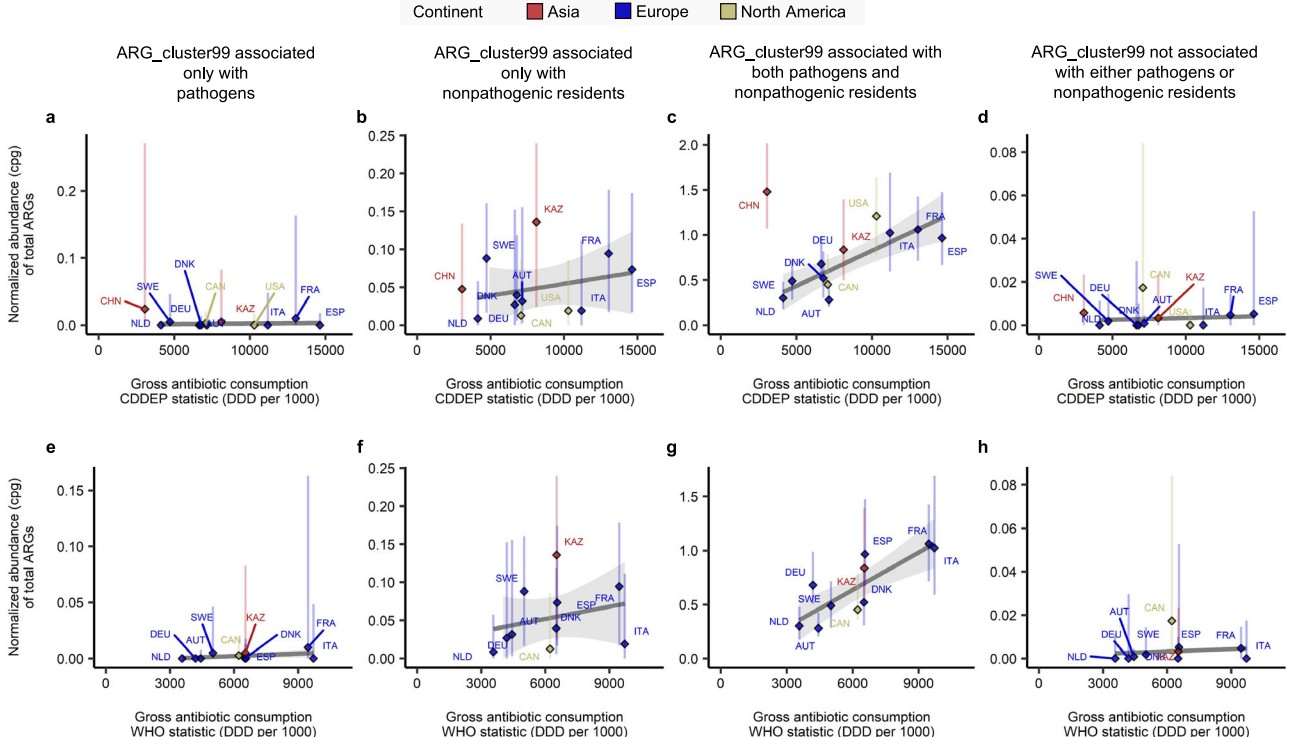

**Fig. 5 | Correlation between median total abundance of ARGs deriving from different species types and antibiotic consumption rates.** ARG_cluster99s were assigned to four categories based on the presence of ORFs from pathogens (see definition in text) and non-pathogenic residents (non-pathogenic found in at least 10% of the stool metagenomes of healthy adults not taking antibiotics). **a, b, c, d** We display correlations between the median ARG abundance (cpg) summed for each species category and the per capita rate of antibiotic consumption (DDD per 1000) in each country using CDDEP statistics. Pearson's correlation **a** r = 0.19, p = 0.58; **b** r = 0.25, p = 0.46; **c** r = 0.81, p = 2.6e-3; **d** r = 0.11, p = 0.76. In **e, f, g, h** we used the WHO antibiotic consumption. Pearson's correlation **e** r = 0.44 p = 0.2; **f** r = 0.26, p = 0.61; **g** r = 0.81, p = 4.2e-3; **h** r = 0.14, p = 0.69. Diamonds give the medians of the

countries. Vertical lines indicate the range from the first quartile to the third quartile. Linear trend line was determined by generalized linear model using ggplot2 R package, shaded area represents 95% confidence interval. The number of metagenome samples used to derive the median and the range bar shown for each country in A-H: AUT (n = 16), CAN (n = 36), CHN (n = 340), DEU (n = 103), DNK (n = 401), ESP (n = 139), FRA (n = 62), ITA (n = 33), KAZ (n = 168), NLD (n = 470), SWE (n = 109), USA (n = 147). Correlation test statistics are given in Table S4. The numbers summarizing two-way categorization of species are provided in Table S9. Comparisons between the results from detection schemes of plasmid-borne ARGs and multi-species ARGs, and between the results from analyzing gut metagenomic ORFs and RefSeq genomic ORFs are provided in Fig. S10.

rates than the peripheral modules (6.1% vs. 0.2%, Chi-sq. test, $\chi^2(1) = 55.82$, p = 8.0e − 14 and Fig. 7d).

## Discussion

We conducted an extensive and geographically wide-spread survey of ARGs in the human microbiome. We resolved substantial previously uncharacterized ARG diversity, with the observed ARG richness varying across body sites. We demonstrated that this ARG diversity is, however, still only a fraction of that present, even for the best-studied body site the gut and therefore, could represent an important potential reservoir of ARGs. We focused on close homologs of known resistance proteins in cultured isolates. This has the advantage that the diversity we do observe is of potentially high clinical relevance. There

are other classes of AMR-conferring genomic features most notably point mutations that provide resistance. We excluded these as comprehensive databases of such mutations only exist for a handful of pathogens, whereas we wanted to survey resistance in as an unbiased fashion as possible. We will also miss ARGs with low sequence similarity to known ARGs. Taking these factors into account the true ARG diversity in the human microbiome could be larger still.

Focussing on the gut microbiome, we observed two distinct phenomena. The first, observed in healthy individuals not currently taking antibiotics, was a substantial difference in both median total ARG abundance (five-fold) and richness (four-fold) across countries. These differences could be largely explained by differences in per capita consumption rates of antibiotics. The strength of the

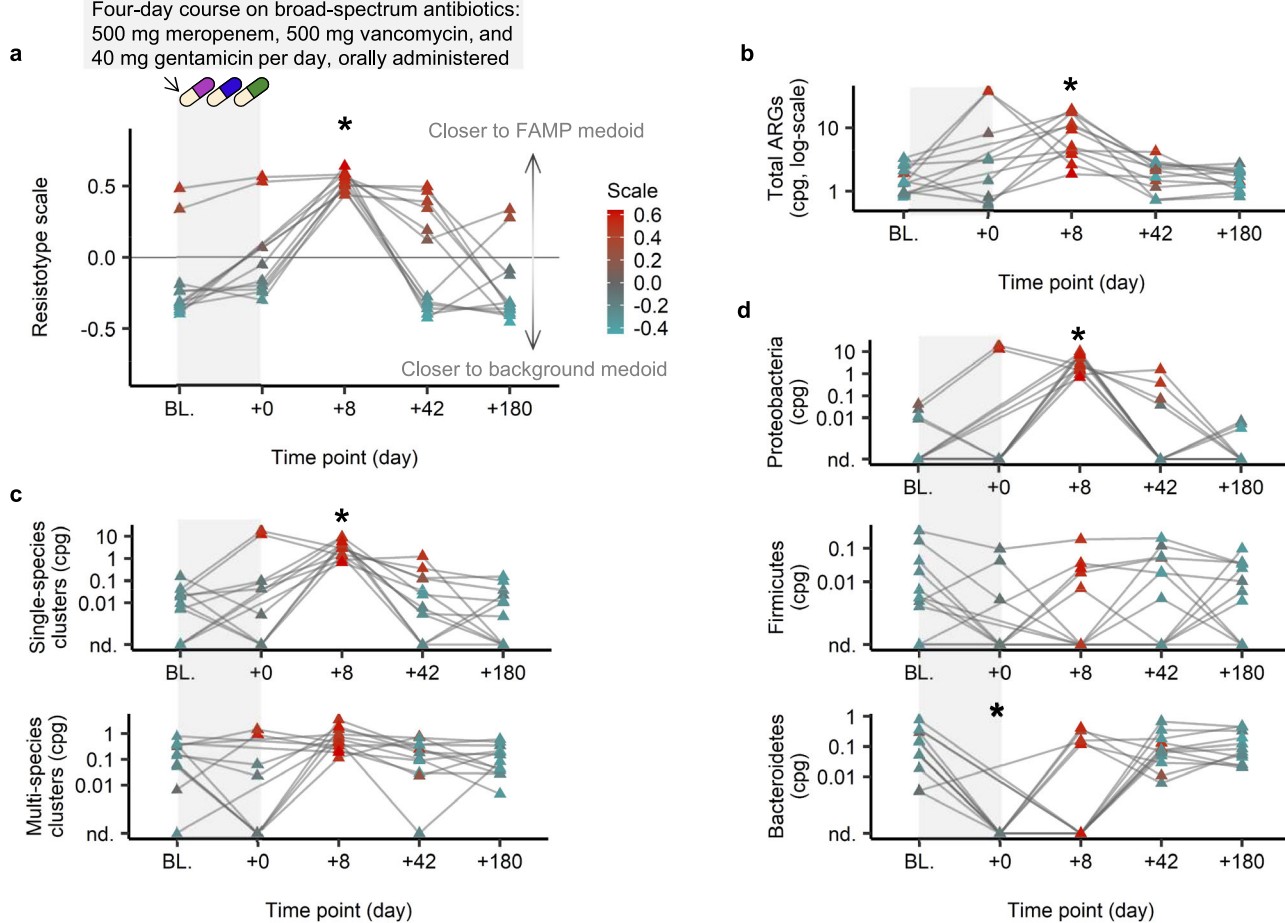

**Fig. 6 | Impact of short-term antibiotic consumption on resistotype assignment and total ARG abundance.** We present individual gut resistome trajectories during a course of antibiotic treatment in healthy individuals[11]. The subjects ($n = 12$) were orally administered 500 mg/day meropenem, 500 mg/day vancomycin, and 40 mg/day gentamicin for four days. **a** Resistotype scale index. **b** Total ARG abundance in normalized copies-per-genome (cpg). **c** Total ARG abundance in single-species or multi-species clusters (cpg). **d** Total ARG abundance across phyla based on taxonomic LCA (cpg). Resistotype scale index (**a**) and total ARG abundance (**b**) were calculated directly using our read-based approach. The ARG abundances stratified by cluster type (**c**) or taxonomic classification (**d**) were derived from the assembly-based analysis (see text). Data points were colored according to the resistotype scale (all panels). Significance compared to baseline computed with the two-sided paired Wilcoxon rank sum test, * Benjamini–Hochberg $p < 0.05$. Exact adjusted $p$-values for the significant case: resistotype scale at day 8, adjusted $p = 0.0020$ (**a**); total cpg at day 8, adjusted $p = 0.0020$ (**b**); single-species clusters cpg at day 8, adjusted $p = 0.0038$ (**c**); Proteobacteria ARGs cpg at day 8, adjusted $p = 0.0038$ (**d**); Bacteroidetes ARGs cpg at day 0, adjusted $p = 0.039$ (**d**). Samples from the same individual are connected by a gray line. Variation of resistotype frequencies by health condition, or by antibiotic exposure status, is provided in Fig. S11.

associations was quite remarkable with correlation values above 0.8 for total ARG abundance. Previous studies have shown a higher abundance of ARGs in microbiomes from individuals deriving from countries with higher antibiotic consumption[21,24], but we have demonstrated a direct correlation between consumption rates and ARG levels in the microbiome at a global scale for healthy antibiotic free individuals. This is in contrast to global wastewater metagenome surveys of ARGs, which failed to find a clear link between antibiotic consumption and ARG abundance[44,45]. This highlights the importance of sampling microbiomes directly from individuals rather than from waste streams where additional factors may be impacting abundance.

When we separated ARGs by the antibiotic classes they provide resistance to, we only found a significant correlation between abundance and consumption for the beta-lactams. This is probably because the beta-lactams are the most commonly used antibiotic class across the countries we studied, representing a mean of 50.1% (WHO data) or 61.6% (CDDEP data) of DDDs, used much more frequently than the two next most common, macrolides at 14.5% (WHO) or 11.6% (CDDEP), and fluoroquinolones with 10.0% (WHO) or 9.8% (CDDEP). This result had a FDR of 0.13 so it should be treated with caution but it suggests that this phenomena is not only operating at the level of overall consumption.

The three countries with the highest abundance and diversity of ARGs were Spain, France and China. For Spain and France these high levels can be explained by their high antibiotic consumption rates. China by contrast, is an obvious outlier in Fig. 2, with an abnormally high level of resistance given its reported antibiotic consumption. There are many potential explanations for this. The samples from China may be atypical in some way, they could derive from unusual localities within China or despite our selection for healthy controls have unreported diseases. However, four separate studies were included in the healthy adult stool samples from China, so we might expect any such biases to average out. Furthermore, this does match with earlier studies that observed anomalously high levels of ARGs in China[21–23]. If these samples are indeed representative of the Chinese population then it may indicate that antibiotic consumption in China is substantially under-reported, or it may reflect an unusually high impact of antibiotics used in agriculture on the human microbiome, most probably from livestock production where China has the highest estimated usage globally[46]. The country with the lowest observed ARG total abundance was the Netherlands, this was consistent with the low levels of antibiotic usage in that country, but the contrast with France, which is geographically close, emphasises the spatial localisation of

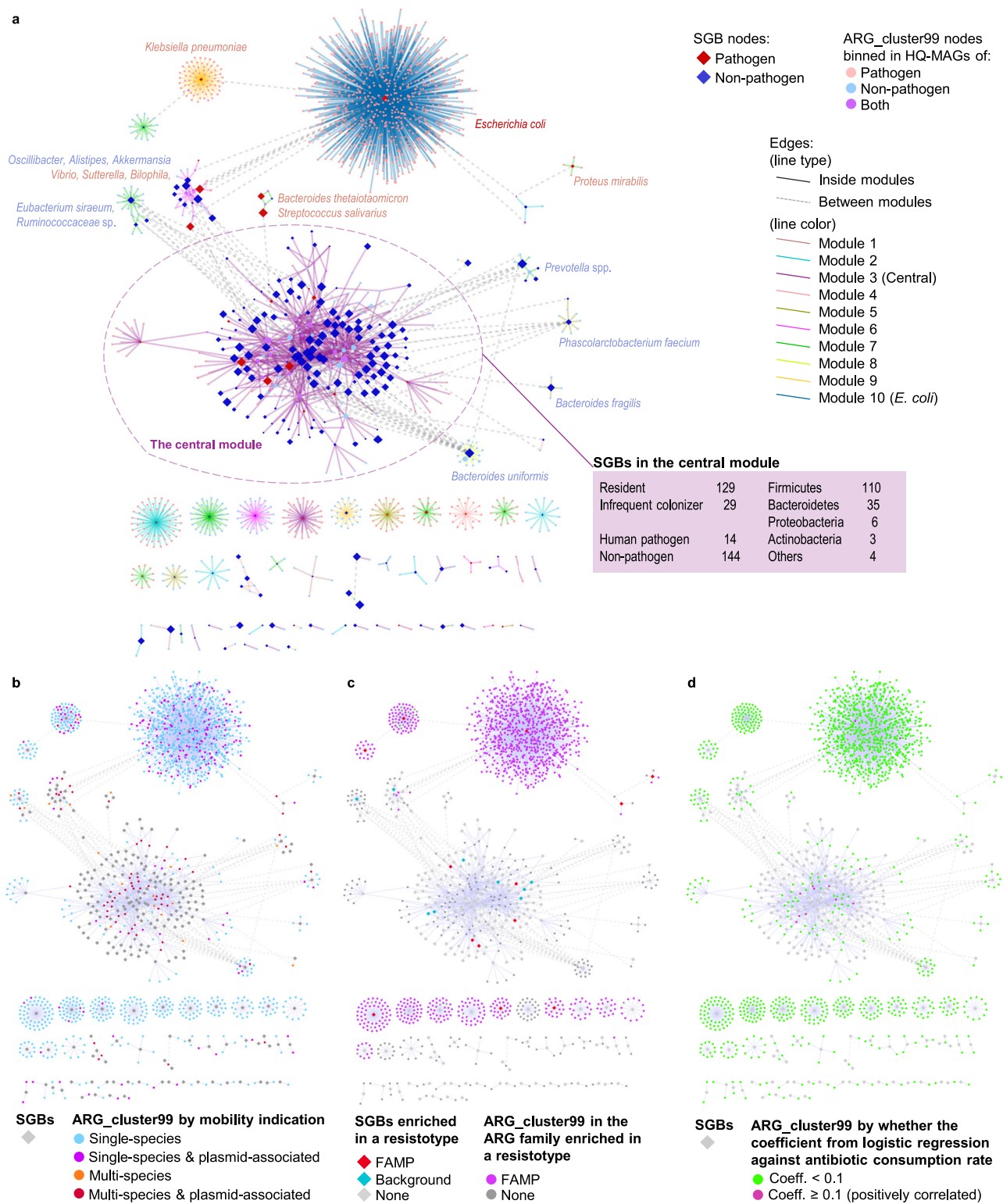

**Fig. 7 | Bipartite network of ARG clusters and species-level genomic bins based on the genomes reconstructed from stool metagenomes.** Network comprising species-level genomic bins (SGBs) connected to ARG_cluster99s they contain (ORF frequency > 1) generated from high-quality MAGs (HQ-MAGs) constructed from 6104 adult stool metagenomes. Ten modules in the network were detected using Beckett's method (computeModules of the R package bipartite). The largest (SGBs *n* = 158) was defined as `central' the others all (*n* < 20) as peripheral. **a** SGB and ARG_cluster99 nodes coloured by pathogen status, edges by module (when shared). **b** SGB not coloured, ARG_cluster99s by single-species, multi-species, plasmid-borne. **c** SGB nodes and ARG_cluster99s were categorised into background- or FAMP-associated. **d** SGB nodes were not categorised, ARG_cluster99s categorised by association with country-level antibiotic consumption (see Methods).

ARGs in the microbiome and the importance of the local antibiotic usage regime.

The second phenomenon, was a clear separation of the ARG family profiles into two clusters or resistotypes. These were robust to the choice of clustering algorithm or distance metric. The less frequent FAMP resistotype was associated with resistance to fluoroquinolone, fosfomycin, aminoglycoside, and peptide antibiotics, as well as multi-drug resistance genes, and a much higher level of resistance overall. The FAMP resistotype was not correlated with antibiotic consumption across countries but was strongly correlated with diseases, particularly enteric infections. We also observed that the short-term consumption of antibiotics causes a rapid shift to the FAMP resistotype. It is possible therefore, that the FAMP resistotype is simply due to individuals that have recently taken antibiotics but we believe this not to be the case, because even when restricted to samples from individuals recruited with a 3 month antibiotic exclusion criterion, we still observed that 22% of gut microbiomes derive from this resistotype. In addition, if this was the case, then we would expect to see a correlation at the country-level with consumption rates, which we do not. We believe that the FAMP resistotype represents a genuine population-level impact of antibiotic consumption relevant to both healthy and diseased individuals.

Methods based on structural similarity such as the study of Ruppé et al. may be more sensitive than the homology searches used here but may also have more false positives[47]. This may explain why in Ruppé et al. six resistotypes were observed rather than two, although that might also be due to the use of a different clustering algorithm, Dirichlet-multinomial mixtures[48], which are not suitable for continuous metagenome ARG profiles.

We were able to attribute these two phenomena to different components of the resistome. The FAMP resistotype is principally driven by single-species ARGs associated with pathogens within the *Proteobacteria*, although some multi-species ARGs shared between pathogens and commensals are also important. In contrast, mobile multi-species ARGs shared between pathogens and commensals dominate the correlations between ARG abundance and consumption of antibiotics at a country-scale. This was confirmed through the construction of a network of species and ARGs. The ARGs driving the population-level response to antibiotics were found in a highly connected central module enriched for resident non-pathogens and mobile resistance genes, with the network periphery which comprises more non-resident pathogens associated with the FAMP resistotype. As part of this analysis, we also quantified the importance of different MGEs in mobilising ARGs in the gut microbiome, with plasmids associating with nearly three times as many ARGs as the next most important, conjugation elements.

We should add two important caveats to the above conclusions, firstly our definition of a pathogenic species, as one with a strain reported to have caused infection at any body site, is imperfect and probably overly broad. Therefore, the ARGs may actually be carried on non-pathogenic strains of opportunist pathogen species. However, the associations we observe, suggest that in a statistical sense our definition is useful and no better definition was apparent to us. Secondly, metagenomics can only determine relative changes in abundance so for example in the FAMP resistotype it is possible that the absolute abundance of resistant pathogens is not higher, rather that the susceptible commensals have decreased, this would motivate revisiting these observations with methods for quantifying absolute microbial loads[49].

The correlations between total ARG abundance and country-level consumption rates were restricted to multi-species mobilised genes but the correlations in ARG diversity were not. An explanation for this may be that the abundance of single-species ARGs are constrained by the ecology of the gut microbiome in healthy individuals and hence a response in abundance will be restricted to multi-species genes that can spread through the community. In contrast, antibiotic exposure can drive an increase in the richness of single-species ARGs even if their total abundance is constrained.

Regarding the clinical relevance of these phenomena, the FAMP resistotype is perhaps more immediately concerning than the country-level correlations. The shift to resistant *Enterobacteriaceae* associated with the FAMP resistotype may potentially lead to an increased risk of resistant opportunistic infections whereas, since the central module is dominated by non-pathogenic residents, the actual clinical consequences of the country-level response to antibiotics may appear less significant. However, this is only true in the short-term, there are some pathogens in the central module and the country-level response is driven by mobile genes that are often shared between pathogens and commensals. There is potential, therefore, for these genes to act as a reservoir maintaining and transferring resistance between pathogen and commensal species, with long-term consequences for how effectively changes in antibiotic usage may eliminate resistance in specific pathogenic organisms.

These two phenomena, operating on different parts of the microbiome and over different time-scales, may be connected. A possible explanation for the FAMP resistotype is that the usage of antibiotics drives resistance in principally pathogenic gram-negative bacteria within a country. This is well established at both country-scales and across US states[15,50]. These resistant gram-negatives then enter the gut either through pathogenic blooms or for opportunistic pathogens e.g. E. coli over longer time-scales as more permanent commensal residents. This explains the FAMP resistotype phenomenon being driven by proteobacterial pathogen associated ARGs and why the FAMP is associated with enteric infections and on a short time-scale by antibiotic consumption.

We then further hypothesise that this pool of resistant organisms associated with the FAMP may transfer ARGs to commensals within the gut microbiome, but that selection for resident commensals with these shared ARGs occurs over a longer time-scale, the degree of selection being dependent on the rate of overall antibiotic consumption in the population as a whole. This explains the overall ARG abundance correlations with antibiotic consumption that we observe.

The above hypothesis is similar to the concept of resistance 'spillover' from individuals exposed to antibiotics to other members of a population[50] but at a whole community level through horizontal gene transfer, with the microbiome acting as a reservoir mediating this spillover, and dysbiosis and a community dominated by *Enterobacteriaceae* as a mechanism accelerating it. As we discussed in the Introduction, there is now good evidence from longitudinal sampling of travellers[16–18], that resistant strains can transfer into an individual's microbiomes from the wider host population, it therefore makes sense to view hosts as embedded within a population-level resistome, that is impacted by overall population-level behaviours. This is just a hypothesis, and there are alternative explanations consistent with our observations, but we hope that the strong population-level impacts of antibiotic consumption on resistance in the human microbiome that we have observed, will serve as a catalyst for further, more mechanistic research in this area.

## Methods
### Metagenome assembly data
The data set consists of 9251 human microbiome samples from multiple body sites (7718 stool, 783 oral cavity, 410 skin, 150 airway, 93 nasal cavity, 88 vagina, and 9 milk) which had been sequenced in various studies (Supplementary Data 2). The assembly data from Pasolli et. al. (2019) includes the depth of coverage for each contig in the header lines, which provided the basis for quantitative profiling of the genes annotated on the contigs. We assessed the integrity of each sample's metagenome assembly based on the recovery rate of the homologs of 40 single-copy genes (SCGs)[51] and removed any samples for which we failed to detect one or more ORF homologs for all 40

SCGs - as those assemblies likely do not provide ≥1x-genome-equivalent coverage. The refined dataset contained 8972 samples including 7589 stool metagenomes of which 6104 derived from adults Table S1.

## Metagenome-assembled genomes and NCBI RefSeq prokaryotic genomes

We combined prokaryotic genome sequences from two resources to create a panel of reference genomes across which we determined the phylogenetic and taxonomic distribution of ARGs. The first data set consisted of 154,723 metagenome-assembled genomes reconstructed from the same human microbiome samples that we obtained metagenome assemblies for[26]. Of these reconstructed genomes 70,178 were labeled as 'high-quality' in the original study, based on >90% completeness and < 0.5% strain heterogeneity. The second data set consists of 152,497 bacterial and archaeal genomes from NCBI RefSeq accessed on 19 April 2019. These genomes included representatives from the principal phyla found in the human gut microbiome although were dominated by *Proteobacteria* (*Proteobacteria*: 83,445; *Firmicutes*: 44,484; *Actinobacteria*: 16,529 and *Bacteroidetes*: 3563, Others: 3634).

Genome sequences from the two sources were clustered into species-level bins (SGBs) based on 5% average nucleotide identity (ANI) radius according to the method described in Pasolli et al. (2019). The list of reconstructed genomes used in this study and their mapping to SGBs and full-rank taxonomy are provided in Supplementary Data 3. The list of RefSeq genome accession numbers used in this study and their mapping to SGBs and full-rank taxonomy are provided in Supplementary Data 4.

## ARG database

We refined the CARD database October 2017 version[29] with the objectives of minimizing false identification of non-ARG specific homologs and maximizing the consistency in ARG family annotation when performing homology search with 80% identity threshold. Starting with the proteins included in the 'protein homolog model' of the CARD ontology, we first removed any housekeeping genes or regulatory genes, the homologs of which may not always imply antibiotic resistance. Next, we clustered the reference proteins by running cd-hit v4.6[52] with 80% global identity threshold. Superclusters of the 80% identity clusters were defined by agglomerative single-linkage clustering of the cluster-representative sequences based on all-against-all blastp. We then built a phylogenetic tree for each supercluster using Muscle v3.8[53] and FastTree2[54].

The protein sequences were assigned to ARG families based on the following rules. (1) If the proteins in a cd-hit-derived cluster share the same gene name prefix, the cluster was finalized as is with the consensus part of their gene names. (2) If a cd-hit-derived cluster contained two or more independent gene name prefixes, we inspected the phylogenetic tree to check if each gene name prefix formed a monophyletic branch. When every gene name prefix could be separated from each other in monophyletic groups, we split the cluster accordingly, otherwise we maintained the original cd-hit cluster and created a concatenated name. (3) If clusters were nested within one another, or intermixed in the phylogenetic tree, or if the clusters sharing identical gene name prefixes appeared as sister clades with marginal divergence, we collapsed the clusters into a single ARG family. This manual curation resulted in the definition of 752 ARG families mapped to the 2159 reference protein sequences.

## Annotation of ARGs and SCGs

Open reading frames (ORFs) of protein-coding genes were defined on the contigs, whether originating from metagenome assemblies or RefSeq genomes, using Prodigal v2.6.2 with '-p meta' option[55]. Protein sequences of the ORFs were searched against two protein databases: (1) clusters of orthologous groups (COGs) the 2014 update version[56] and (2) a manually refined version of the CARD as described above. The

query (meta)genomic ORFs were aligned to the database proteins using the blastp function of Diamond v0.9.9[57]. COG numbers were annotated to the ORFs according to the reference COG protein displaying the highest bit score after filtering the hits by maximum e-value 1e-7. ARG family names were annotated to the ORFs according to the reference CARD protein displaying the highest bit score after filtering the hits by maximum e-value 1e-20, minimum identity 80% and minimum reference coverage 80%.

## Calculation of normalized abundance of ARGs in metagenomes

We used an assembly-based approach, in which we a calculate normalized abundance value for each ARG annotated ORF found on an assembled contig, in the subset of analyses where we stratified the ARG abundance profile according to the taxonomic or phylogenetic distribution and MGE context associated with the ARGs. On the other hand, for the purpose of simply profiling at the level of ARG families (*i.e.*, clusters of ARGs defined in the reference database described above) we used a more sensitive read-based approach.

In the assembly-based approach, we assigned coverage depth value to each metagenomic ORF using the coverage depth of the corresponding contig. Raw unnormalized abundance value was calculated for each ARG family in a sample as the summed coverage depth values of all ORFs that were annotated to that ARG family in the given sample. In the same way an unnormalized abundance value was calculated for each of the 40 SCGs. We divided each ARG family's unnormalized abundance value by the median of the unnormalized abundance values across 40 SCGs to give a normalized abundance that is equivalent to 'copies per genome (cpg)' - in a sense that the normalized value 1 would mean that the summed depth of the ORFs annotated to the given ARG family across the metagenome contigs would be the same as that of a typical SCG.

In the read-based approach, we first aligned raw reads of the sample against the reference proteins in our curated version of CARD, using 'diamond blastx -k 1 -f 6 -e 1.e-10 --id 80 --query-cover 70' using Diamond version v0.9.9[57]. Then the blastx alignments were grouped by the reference ARG family, each ARG family was subsequently inspected for the coverage breadth (the number of length/100 intervals that were covered at least once) by the collection of blastx alignments. Any ARG family displaying coverage breadth less than 80 was discarded, which is equivalent to the 80% coverage threshold that we applied in the assembly-based approach. Cpg values were calculated for each ARG families that passed the 80% coverage breadth by dividing the ARG family's reads-per-kilobase (RPK) by the median RPK of the 40 SCGs.

We compared the ability of assembly-based and read-based approaches to detect and quantify ARG families in the stool metagenomes based on 5341 adult stool samples that we processed using both methods.

## Clustering of ARG sequences

Nucleotide sequences of the metagenomic ORFs and the genomic ORFs annotated as ARGs were pooled together and clustered using four different thresholds to create the clustered catalogues of ARGs. We ran the 'cluster' command of MMseqs2[58] using '--min-seq-id 0.9 -c 0.8' (for ARG_cluster90), '--min-seq-id 0.95 -c 0.8' (for ARG_cluster95), '--min-seq-id 0.99 -c 0.9' (for ARG_cluster99), and '--min-seq-id 1.0 -c 0.9' (for ARG_cluster100), under '--cov-mode 0'.

## LCA taxa assignment and identification of multi-species ARGs

We determined the lowest common ancestor (LCA) taxon for each ARG cluster using the application galaxy-tool-lca when the cluster contained at least one ORF derived from a HQ-MAG or RefSeq genome. SGB assignments on the member ORF-affiliated MAG or RefSeq genome were used as the input taxa for the LCA calculation, along with the full-rank taxonomy provided by the Pasolli et al. 2019 method.

Based on the LCA assignments, we determined which ARG_cluster99s had evidence of recent horizontal gene transfer, by classifying them into one of the following categories: species-specific when the rank of LCA is at the species level, multi-species when the rank is higher than species, and LCA-unassigned. The latter occurs when none of the ORFs belonging to the cluster has known SGB affiliations. We propose that the multi-species ARG_cluster99s with ORFs deriving from different species may have experienced recent HGT and hence are mobilised.

We validated this approach by clustering a panel of 40 SCGs at a range of nucleotide identities. These SCGs which are often core housekeeping genes will be in general vertically transmitted albeit more slowly evolving than accessory genes. This is confirmed by the fact that the majority of SCGs coalesced into multi-species clusters at cut-offs around 97%-98% below our 99% threshold for horizontal transfer.

In Supplementary Data 6 we give binning and multi-species rates for all ARG families.

## Rarefied richness of ARG clusters

We used a random subsampling strategy to compare the diversity of ARG clusters between body sites and countries whilst adjusting for the differences in metagenome sequencing depth across samples. To construct rarefaction curves giving the number of observed ARG clusters as a function of sample number or size for each body site, we first created a matrix giving presence/absence of the clusters (columns) in the samples (rows) per each body site. We then collected random subsets of $q$ samples with $q$ increasing by 10 until it exceeded the total number of samples in the body site. The process was repeated 99 times. For each random subset we recorded (1) the number of subsamples, $q$, (2) the total sum of sequence (Gbp) in subsamples, and (3) the number of clusters that were present in at least one subsample, our measure of rarefied ARG diversity. To compare ARG richness across the countries we found subsamples for each country with a total of $100 \pm 10$ Gbp of sequence data. In detail, we first created a matrix of presence/absence of the clusters in the country's samples and repetitively performed incremental subsampling where the number of subsamples $q$ started from 1 and increased by 1 until the accumulated total amount of sequence in the subsample exceeded 110 Gbp. After each round of incremental sampling finished, we determined $q'$, the sample number that produced the accumulated sequence data size closest to 100 Gbp. If this was within 90 - 110 Gbp, we recorded (1) the number of subsamples, $q'$, (2) total amount of metagenome sequence, and (3) the number of unique clusters observed in the subsamples, the rarefied diversity. The process was repeated for each country until 99 successful replicates, *i.e.* with $100 \pm 10$ Gbp of sequence data, were generated.

## Machine learning-based identification of plasmid sequences

We adapted an existing machine learning approach, PhageBoost, developed to identify prophage sequences[30], to the problem of classifying plasmid sequences. The resulting program, PlasmidNet, uses the same principles as PhageBoost, but was retrained using a custom database of RefSeq sequences identified unambiguously as genomes and plasmids. It uses the same features from ORFs as PhageBoost e.g. ORF length, amino acid dipeptide and tripeptide frequencies but the underlying XGBoost machine learning algorithm was adapted to the TabNet deep tabular data learning architecture[59]. We first classified ORFs as deriving from plasmids or genomes based on this model and then assigned contigs according to their ORF consensus. PlasmidNet on a hold-out test data set of 11784 genome contigs and 14223 plasmid contigs achieves a false negative rate of 1.4% with a false positive rate of 15.5%. We ran PlasmidNet on the entire set of contigs from the metagenome assemblies and RefSeq genomes analyzed in our study.

PlasmidNet is available from: https://github.com/kkpsiren/PlasmidNet.

## Annotation of MGEs

We annotated the hallmark proteins of class 1-4 integrons, IS elements, and conjugative systems to determine the MGE contexts other than plasmid around the ARGs. All contigs from metagenomic assemblies and RefSeq genomes were subjected to this analysis. For integrons we used integrase proteins (IntI) as the hallmark. We searched ORFs against a database containing six sequences, the representative sequences of IntI1-4 (AAQ16665.1, AAT72891.1, AAO32355.1, and 99031763) and the two outgroup sequences (P0A8P6.1 and P0A8P8.1), using diamond blastp (version 2.0.13) with –id 80 –subject-cover 80. For IS elements we used IS-associated transposases as the hallmark. We searched ORFs against the transposases retrieved from ISFinder[60] using diamond blastp (version 2.0.13) with –id 80 –subject-cover 80. For the conjugative elements we used the hallmark proteins provided by the CONJscan database[61]. We searched for alignments to all modules (e.g., CONJ, MOB, typeB, typeC, etc.) included in the CONJscan HMMs, using hmmsearch (version 3.2.1) with score threshold (-T) of 40. For each and every ARG ORF annotated on the genomic or metagenomic contigs, we determined distances to the closest IntI, IS transposases, and conjugative system proteins, respectively. The ARGs found within 100 Kbp of IS transposases or conjugative system proteins were assigned to be putatively associated with those MGEs, since the size of known composite transposons range up to 80 Kbp[62] and the size of conjugative mobile elements up to 100 Kbp[63]. The ARGs found within 10 Kbp from IntI were assigned to be associated with integrons, since the majority of integrons in bacterial genomes have 10 Kbp or shorter cassette array length[64].

## Species-level profiling of metagenomes using SCGs

We first created a non-redundant reference sequence database complete with taxonomies for each of the 40 SCGs. To do this, we annotated COGs on the RefSeq genomes and the HQ MAGs. We selected each set of SCG ORF sequences. The collected sequences were clustered using the 'linclust' command of MMseqs2[58] at 100% identity and 90% sequence overlap using the parameters '–min-seq-id 1.0 -c 0.9 –cov-mode 1' and each cluster was then assigned the taxonomy of the SGB that the majority of ORFs derived from.

To profile the SGB compositions (*i.e.* species-level community compositions) of the microbiome samples, we collected nucleotide sequences of the metagenomic ORFs that were annotated to the 40 SCGs and searched them against the corresponding reference SCG sequence database, as described above, using vsearch v2.4.3[65] with parameters '–id 0.9 –query_cov 0.5'. The metagenomic ORFs were assigned the SGB of their top-hits in the database, the coverage depths of all ORFs assigned to an SGB were then summed to obtain a vector of SGB abundances for each sample. The coverage abundances were normalised within each sample to have a sum of 1 to give a compositional matrix of SGBs. We excluded samples which started off with less than 1000 SCG-annotated ORFs from the subsequent analysis of variation in species compositions.

## Sample metadata and selected data subset for population-level gut resistome analysis

Organized sample metadata were obtained from the curatedMetagenomeData R package[25]. For the population-level gut resistome analysis, we focused on stool samples and controlled the host age by excluding 1435 stool samples from the 'children' or 'newborn' categories from the 'AgeCategory' column in the curatedMetagenomeData table. The remaining 6112 samples were again filtered by the country of origin by excluding countries represented with less than 10 samples. The resulting set of 6104 samples comprised adult (defined here as post-childhood) gut microbiomes from 20 countries Table S1. For the adult gut microbiome samples, we conducted additional metadata collection from the corresponding literature and via inquiries to the original data producers to fill in missing

information related to host health (disease) status and recent antibiotic use. As a result, we were able to identify unambiguously 3565 samples as deriving from the gut microbiome of healthy individuals. Samples excluded from the 'healthy' category include (i) 1658 samples with various disease labels (CDI, cholera, colorectal adenoma, CRC, IBD, liver cirrhosis, T2D, fatty liver, hypertension, rheumatoid arthritis, and STEC), (ii) 750 samples without any information regarding the host health and disease status, (iii) 58 healthy controls that clearly derive from non-contemporary subjects (i.e. collected in 1980s) and (iv) 73 samples from populations maintaining traditional lifestyle (i.e. hunter-gatherer). It is still possible though that some of these 'healthy' individuals have undocumented diseases irrelevant to the original studies.

Using a combination of sample information from curatedMetagenomeData and the original study protocols we identified 4155 adult stool samples the donors of which were unambiguously not taking antibiotics at the time of collection. This includes cohorts with any guaranteed length of antibiotic-free periods preceding the sampling as declared in the recruitment criteria. In the remaining samples, 273 were 'currently' on antibiotics and 1676 had no available information. Among the 3565 healthy adult samples, 3096 were identifiable as not taking antibiotics at the time of sample collection.

### Country-level statistics for antibiotic consumption
National antibiotic consumption data was collected from the ResistanceMap operated by the Center for Disease Dynamics, Economics and Policy (CDDEP)[34] and the World Health Organization (WHO) report[35]. CDDEP provides antibiotic consumption rates in the units of defined daily dose (DDD) per 1000 (capita) throughout the corresponding year, while WHO provides DDD per 1000 per day. We converted the CDDEP data to DDD per 1000 (capita) per year – (DDD) per 1000 – by multiplying by a factor of 365 so that the two data sets had equivalent units. As the national data in the WHO report mostly comes from surveillance made in 2015 and the CDDEP provides data for the years from 2000 to 2015, we used the data from the year 2015 in our correlation analysis. Note that from the CDDEP data we found that the ranks of the countries analyzed in this study in terms of DDD per 1000 do not change over the years.

### Definition of human pathogenic species
We compiled a list of species names of bacteria that have been reported as the causative agents of human infectious diseases, through manual review of a clinical microbiology manual[42]. The compiled list contains 463 species names and is available at https://github.com/kihyunee/gut_resistotype. We matched SGBs to those pathogen names by comparing the NCBI taxid attached to the pathogen names and to the RefSeq accession numbers. When more than one SGB matched to a single pathogen name, the pathogen name was assigned to the SGB with the largest number of RefSeq entries that are linked to the taxid. A table assigning pathogen names to the SGBs are provided in Supplementary Data 5. Virulence factors were annotated on the set of RefSeq genomes and MAGs using VFDB set A (as of 07-August 2022)[66] as a reference database and 90% identity over 80% subject coverage as thresholds for diamond blastp. We categorised RefSeq genomes into clinical and non-clinical isolates based on the 'epi_type' field in the isolate metadata tables obtained on 28th July 2022 from NCBI Pathogen Detection https://www.ncbi.nlm.nih.gov/pathogens/.

### Quantification and statistical analysis
**Correlations tests between ARG abundance and diversity with antibiotic consumption rates.** We tested for correlation between the median abundance of ARGs (cpg) in the gut microbiome of healthy adults who were not taking antibiotics and the national antibiotic consumption rates (DDD per 1000). Countries were included in the test only if at least 10 subjects were available. In addition to the total

abundance of ARGs, various subcategories of ARGs were subjected to this test, including the ARG_cluster99s that are multi-species (mobile), species-specific, and LCA-unassigned, and the plasmid-borne ARGs and non-plasmid-borne ARGs. We tested using CDDEP and WHO per capita consumption rate data separately. For each correlation test, we first tested the normality of country-level median ARG abundances using a Shapiro-Wilk test, we then applied Pearson's correlation if the medians showed a normal distribution, otherwise a non-parametric Kendall's correlation was used. We also tested the correlation between the country-level ARG cluster richness estimates (median across 99 iterations of 100 Gbp-targeted subsampling) and the antibiotic consumption rates, using the same statistical method. All tests were performed with R core functions.

**Negative binomial regression of number of shared clusters as a function of SGB phylogenetic distance and co-occurrence.** For each species SGB we determined the total number of unique ARG_cluster99s observed across all MAGs and reference genomes assigned to that species. We then considered every pair of species and counted the number of clusters that were shared between them. The maximum number of shared clusters that could be observed between two species is the minimum of the number of clusters associated with each. We performed a negative binomial regression (using the glm.nb function from the MASS package of R) of number of shared clusters using the maximum as an offset term to effectively predict rate of cluster sharing as a function of both phylogenetic distance between the pair and the fraction of sites they co-occur in.

**Multivariate statistical analysis of ARG profiles.** We identified outliers using the assembly-based family coverages. First, we prepared the abundance (cpg) matrix of 6,104 adult gut samples (rows) and 752 ARG families (columns). Next, we removed samples with less than three ARG families (remaining $n = 6022$) and the ARG families with zero occurrence (remaining $n = 363$). The remaining matrix was log-transformed after adding 0.1 * (minimum non-zero cpg value in the matrix). Non-metric multidimensional scaling (NMDS) was performed with the metaMDS function of the 'vegan' package and Euclidean distances. To remove outlier data points, we calculated the Euclidean distance from each sample to the global medians of MDS1 and MDS2, and then iteratively ran Grubbs's test implemented in the 'outlier' R package on the vector of Euclidean distances, removing outlier samples ($p$ value cutoff 0.05) until no more outliers were detected. This resulted in a set of 6006 adult gut microbiome samples.

For the resistotype analysis presented in the paper, we used the read-based profiles. We could obtain sequence reads for 5469 adult gut samples. As above, we removed samples with less than three ARG families (remaining $n = 5457$) and the ARG families with zero occurrence (remaining $n = 422$). Among these 5457 samples, we selected those that were not-outliers based on the assembly-based profiles, *i.e.* they were included in the 6006 samples above, that gave 5372 samples remaining in the read-based cpg matrix.

The final sample set was again inspected by NMDS, as well as by Uniform Manifold Approximation and Projection (UMAP) using the 'umap' package. Partitioning of the ARG profiles into resistotypes was assessed using the partitioning around medoids (PAM) method implemented in the 'cluster' package. Bray-Curtis distances were used for the resistotype assignments but the PAM clustering was repeated with Euclidean and Manhattan distances to test for consistency in the number of predicted clusters. For the same reason, k-means with the elbow method was also evaluated. Variation in the species compositions of adult gut microbiome samples were inspected with the same overall procedures except that the compositional matrix was used as is, without log-transformation with Bray–Curtis distances.

**Selection of SGBs for construction of the phylogenetic tree.** 522 SGBs were selected for inclusion in the tree of Fig. 4 that are: i) detected in 1% or more of the adult stool metagenome samples, (ii) had at least 10 genomes, (iii) have three or more relevant ARG families, or showed significant association with one resistotype (≥5-fold difference in mean relative abundance and $p < 0.05$ by Mann–Whitney tests after Benjamini–Hochberg adjustment).

**Identification of ARG clusters associated with country-level antibiotic consumption rates.** We selected the gene clusters that occurred in 100 or more samples (clusters $n = 268$) and obtained the relative abundance (cpg) profiles of these clusters throughout the samples from the 15 countries where we had CDDEP statistic for the gross antibiotic consumption rate of the year 2015 (samples $n = 4738$). Logistic regression was performed on the binary, presence or absence, profile of a cluster in a sample against the rank of the countries according to the total antibiotic consumption rates. We defined the gene clusters that showed coefficients of 0.1 or greater as a group of gene clusters that are positively correlated with the country-level antibiotic consumption rates (positively correlated clusters $n = 13$).

### Reporting summary

Further information on research design is available in the Nature Portfolio Reporting Summary linked to this article.

## Data availability

We downloaded metagenome assemblies generated in a previous study[26] fasta files accessed at: http://segatalab.cibio.unitn.it/data/Pasolli_et_al.htmlRaw Illumina sequencing reads of these metagenome samples were downloaded from the read archive at the NCBI or the EMBL using kingfisher: https://github.com/wwood/kingfisher-downloadwhen the run accession number was available. The list of sample identifiers analyzed using raw reads can be found along with the matched run accession numbers in the Supplementary Data 1.

The ARG catalogue and abundance profiles generated in this study and the sample metadata table can be accessed at: https://doi.org/10.5281/zenodo.7188053 Source data are provided with this paper.

## Code availability

The analysis scripts used in this study are available from: https://github.com/kihyunee/gut_resistotype and the exact version used for this analysis archived at: https://doi.org/10.5281/zenodo.7465315

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

## Acknowledgements

This work was supported by the Korea Center for Disease Control and Prevention (2017ER540701). CQ was supported by the BBSRC Core Strategic Programme Grant (BB/CSP1720/1, BBS/E/T/000PR9818, and BBS/E/T/000PR9817). S.R is funded through NERC grant ResPharm (NE/T013230/1). F.H. was supported by European Research Council H2020 StG (erc-stg-948219, EPYC), the Biotechnology and Biological Sciences Research Council (BBSRC) Institute Strategic Programme Gut Microbes and Health BB/r012490/1 and its constituent project BBS/e/F/000Pr10355. This work was also partially supported by the European Research Council (ERC-STG project MetaPG-716575 and ERC-CoG microTOUCH-101045015) to NS.

## Author contributions

K.L. performed the bioinformatics analysis, statistics and generated figures. C.Q. and C.C. supervised analysis. S.R. assisted with bioinformatics pipelines. K.S. wrote the plasmid classification tool. F.A., F.C. and N.S. provided curated MAG collection. F.H. assisted with metadata interpretation. K.L., C.C., and C.Q. principally wrote the MS but all authors contributed.

## Competing interests

The authors declare no competing interests.
