## [Peer Review File · Nature Communications]

REVIEWER COMMENTS

Reviewer #1 (Remarks to the Author):

In this manuscript, Lee et al investigate correlations between population based antibiotic usage and the proportion of ARGs in human gut microbiomes. Finding a positive correlation between these two, they then attempt to link ARGs and MAGs. One overall limitation of this study is the focus on human antibiotic usage. The anomalously high resistance levels in China, for example, might be the result of antibiotic use in other industries (such as the livestock industry). The authors then identify a major and minor resistotype in a nice cluster-based analysis. The minor resistotype is associated with a higher abundance of proteobacteria and more resistance gene types, as one would expect/predict. While the authors took good care to remove samples from subjects who may have been exposed to antibiotics at the time of sampling, I wonder if the minor resistotype reflect subjects who recently were exposed to antibiotics (and thus have an enrichment of proteobacteria that harbor more numerous and diverse ARGs). Overall, I found the manuscript interesting and well written. I am hard pressed to understand and make sense of the fact that the conclusions of this manuscript suggest only two resistotypes whereas the Ruppe findings suggest many more. I wonder to what extent the minor resistotype (which looks a lot like the composition of microbiomes that are exposed to antibiotics) drives the MDS results and suggests only 2 clusters. Put another way, if that minor resistotype is removed and clustering is done again, does one see more subtle differences or sub clustering within the major resistotypes? On balance, this is a small point, but it would add some intrigue to the manuscript. As it stands, an antibiotic usage expert with a microbiology background would probably be unsurprised by the results presented here - that there is a resistotype that is enriched for ARGs and that the ARGs are largely proteobacteria-associated. I am thus wondering if there are ways to expand upon the very nice analysis performed here to provide a reader with more insight - thus increasing the impact of the manuscript. Below, I offer several major and minor comments on ways to potentially accomplish this objective.

Major:

1. One of the limitations of the approach used here, where ARGs are identified from the assemblies as opposed to the reads is that the ARG has to be in a sufficiently well covered genomic region/MGE to be assembled well (usually covered $>5x$). This limitation should be made clear in the text. It would be interesting to compare the assembly based results vs. short read based ARG quantification approaches. For example, the finding of a median of only 14 ARGs per metagenome seems low. How would this compare to a method that uses reads to call ARGs (as opposed to assemblies)? I anticipate the latter would be more sensitive.

2. The authors correctly point out that methods like HiC are required to map plasmid MGEs to chromosomes; alternatively, methods like long read sequencing can incorporate integrated MGEs into chromosomal DNA. Both plasmids and integrated MGEs are underrepresented in binned MAGs; thus the approach outlined to investigate MAGs likely undercalls the proportion of ARGs that are associated with genomes. What proportion of the ARGs, for example, are found in the “unbinned” fraction?

3. The authors do attempt to identify plasmid-associated ARGs; but do not have a specific approach to account for integrated HGT segments that can integrate into genomes and carry ARGs. This is an important source of an error by omission.

4. Is the ARG abundance perhaps related to issues beyond antibiotic usage per capita; For example, the amount of antibiotics used in rearing of livestock, the prevalence and range of antibiotics found in drinking water, etc? While the positive correlation observed is interesting, it is likely that many features beyond antibiotic usage per capita are contributing to this finding.

5. The rarefaction that was performed to account for differences in sample number across body sites was a good idea; one thing that isn't well accounted for is the known variation in skin microbiome composition at different skin sites. For example, the palm of the hand has different microbes from the retroauricular fold. Thus, if all of the skin samples (or most) were from one body site, there still may be undiscovered diversity of ARGs in other skin sites on the body.

6. I am curious to know whether the ARGs that were mappable to taxa were different classes than those that were not mappable to taxa (line 257). For example, I would predict that tetR genes often were mapped to taxa whereas beta lactamases were often not mappable to taxa (as they are more likely to be on MGEs). This analysis might help define which ARGs are more likely to be found on MGEs and thus are also more likely to be transferred between species.

7. Figure 7 - it is interesting to note that the central module contains genes that are frequently shared between organisms in the central module, but that these are almost never shared with pathogens. This suggests that gene transfer is happening between related and unrelated gut microbes (commensal type) but that this isn't necessarily leading to an enrichment of ARGs in pathogens (which seems to be the premise for the study).

8. I am not an expert statistical reviewer, but the analyses performed seem sound and generally well grounded. That said, it would be valuable to seek to comments of an expert statistical reviewer.

9. There are many statements of novelty in the manuscript (esp in the intro and discussion); given this, it would be helpful for the reader to see a supplemental table that provides an overview of the studies of ARGs in the human microbiome to date. Alternatively, if this journal does not allow statements of novelty, perhaps these comments should be removed.

Minor

1. 'Form' should be 'from' - line 96

2. Microbiome is misspelled on line 96

3. Given that only 1 milk sample is used, it would seem reasonable to omit this sample type; or to expand to a larger resource/database that has more milk samples.

4. How was the 80% identity cutoff / 80% target sequence length (line 104) selected?

5. I don't know that we are certain that the gut is the most "metabolically active" part of the microbiome; I might recommend deleting this phrase (line 155).

6. One limitation of excluding infants and children from this analysis is that children are often exposed to antibiotics; and thus might be more likely to harbor ARGs that may be lost by adulthood. I wonder if it might be worth redoing this analysis to include infants/children; or to do an additional separate analysis with this cohort.

7. Line 167 onward - While the healthy controls may not have a diagnosis of a given disease that was studied in the parent study, it is important to note that when aggregating data from other studies, "healthy" is not a term that is uniformly defined across studies. For example, in a cancer study, a subject who has hypertension and hyper lipidemia, but has never had cancer, may be counted as a "healthy control".

8. I am curious about the distribution of the ARG copies per genome - is it unimodal, bimodal, normal or skewed? It would be nice to see these data in a figure or supplemental figure. Some resistance genes (e.g. tetR genes) are often chromosomal and would be expected to be in a single or few copies per genome range. Other ARGs (on mobile elements, etc) might be present in varied copy number (potentially even very high cpg).

9. One thought is that ARGs/metagenome may be higher in more diverse samples or individuals that have a long tail distribution of strains/species. How is this accounted for in the analysis? For example, Individuals in France might have a high number of observed ARG clusters but that may be due to having more diverse taxa (at a more even representation and thus more likely to be assembled into contigs). This is why a read based approach may also be helpful in teasing these relationships apart. I believe that the results presented are likely correct - but these are confounders that should be considered and would be helpful for the reader to see and understand. The cpg

analysis somewhat accounts for this, but assumes that all organisms (or at least most of them) from a given sample have a MAG. This is almost never the case given the standard range of sequencing depth that is typically pursued.

10. Figure 3 - It would be helpful if FAMP was defined in the legend (for readers who will simply scan through the figures and legends).

11. Figure 4 - perhaps I missed this, but what resource was used to categorize “pathogens” vs. “non-pathogens”? I don’t have a particular problem with this designation, but as this is a hot topic issue (using the term pathogen as a binary category) in the microbiology community, it would be helpful to make clear how this categorization is being done. As an example, *Bacteroides fragilis* can be a pathogen, but typically is not and is a common member of gut microbiomes. So would it be categorized as a pathogen or a non-pathogen for the purpose of this analysis?

12. What were the cutoffs for sequencing depth (bp sequenced) for inclusion in this study? Also, what was the range of coverage? The results are sensitive to variations in this. If one country, for example, had samples that were less deeply sequenced than another, one might erroneously conclude that there are fewer ARGs in those metagenomes.

13. I appreciate that PlasmidNet is a modification of the PhageBoost algorithm; to be included here, it would be helpful if PlasmidNet was benchmarked properly and described in terms of its performance characteristics. Else it is hard to know how to evaluate results from this tool.

Reviewer #2 (Remarks to the Author):

The article by Lee et al., analyzed human metagenomes in order to understand how antibiotic resistance genes (ARGs) are distributed among different human cohorts and countries, and what factors drive the patterns such as the usage of antibiotics at the country level. While several of the findings reported such as correlation with antibiotic usage, frequent horizontal transfer of ARGs, duration of the effects of the antibiotic administration on the microbiome, echoed previous findings and thus do not represent novel results, the scale that this study was performed and the results provided in terms of quantification represent novel contributions, to the knowledge of this reviewer. Further, the two types of resistomes reported also represent interesting findings. The paper is overall well written and does not require much editorial editing albeit it is quite long (I made some suggestions about shortening below). That said, there are several methodological and interpretation issues that require more attention by the authors. Most notably, the approach for estimating ARG abundance, the lack of correlation between a specific ARG and the usage of its corresponding antibiotic, and the possible effect of diet or asymptomatic infections on the resistome types and the outlier Chinese samples (see also specific points below). Further, is the minor resistome driven mostly by Chinese samples? Is this likely due to diet e.g., higher frequency of hot spicy food (some spices are related to antibiotics, I think)?

Minor/Specific points

“Proteobacteria” in the abstract and other taxonomic names elsewhere should be in italics.

Lines 63-67. It seems that the justification for the study is not very strong, as stated here. Perhaps the authors could highlight more that the quantitative aspect is lacking in previous publications, and the usefulness of different “states” or “types” of resistomes for antibiotic usage and management. I do think that the study has enough novelty although this is probably undersold here.

Line85-90. These do not represent news really. The authors could focus more on the novel findings reported in their manuscript, which could help reducing the length of the paper and thus, make it easier to read.

Line 113. What "unique" ARGs means here? It is not at 95% nt identity as the previous sentence indicate, right? Also, the previous sentence needs a citation to back up the threshold used.

Lines 142-146. Interesting data and I think the paper has enough novel results and thus, could make a stronger justification around quantification. (Related to the major suggestion above)

Lines 180-185. Is it possible that the higher diversity signal in the Chinese samples is due to more people sampled or in more remote/not connected areas? Or diet and spicy foods?

Line 190-193. There is potentially an issue with how abundance is calculated, and this could possibly underly the lack of signal/correlation between the abundance of specific ARG and the usage of their corresponding antibiotic. That is, to take the abundance of the contig as the abundance of the ARG is potentially problematic because it depends on the coverage (and the nature) of each gene on the contig (i.e., better to estimate abundance on a per ARG gene basis). Further, it would be better to map the reads on ARG sequences with a threshold for identity, and not rely on the assembly because the assembly identity threshold is not defined and could be variable, and lower than the 99% threshold used to defined ARG variants. Calling ARG on contigs seems OK/robust. In the view of this reviewer, it would be helpful for the authors to validate their abundance estimation method and show that any biases did not significantly affect their results and conclusions. This does not have to take place for all genes, which is computationally very expensive, but just for a small subset of genes and samples/metagenomes.

Line 257-233. Or that the abundance estimates are not highly reliable at the individual ARG family?
See related comment just above. This result reported here does not make much sense and the authors should investigate it further, I believe.

Line 325. No comma needed at "In order, to"

Line 407 and elsewhere. I would not call them pathogens by the taxonomy only, especially for enterics; e.g., I would look, in addition, for presence of the known virulence genes. Consider the E. coli case for instance; some E. coli are commensal and do not carry the major virulence factors such as toxins, hemolysins etc. Related to the pathogen work: is it possible that the minor resistome represents bacterial infections, and not healthy states as originally thought? Please double check the metadata for the corresponding samples.

Lines 465-475. I suspect that the authors just repeated the results of the previous study here and thus, this section could be removed (if my suspicion is true).

Network analysis. I am not sure what the authors can infer from this network work; e.g., they can not know which node/population gives the ARG to which. Further, the composition of the network seems like the expected one based on the relative abundances of high vs minor resistome types and the membership of the types. Accordingly, this section could be reduced without much loss in clarity, and the discussion on line 595 and on is more like hand-waving (since direction of horizontal gene transfer can not be established).

Line 532. phenomena, not -non.

Line 540-542. There might be a diet effect as well e.g. more spicy food?

Reviewer #3 (Remarks to the Author):

- In this manuscript, the authors conducted a large survey of antibiotic resistance genes (ARGs) in publicly available metagenome assemblies spanning multiple countries. The authors claim that the

ARG richness in metagenomes positively correlates with the reported antibiotic usage rates in the countries under investigation. The authors claim to have detected networks of mobile ARGs that are transmitted among pathogenic and commensal members of the human microbiome. Furthermore, the resistomes are suggested to group into two distinct clusters: major and minor resistotypes. The minor resistotype is claimed to have higher ARG burden, consist of higher proportion of mobile ARGs, and be overrepresented in pathogenic organisms. While this work analyses the largest dataset of this type, the conclusions often appear confirmatory of previous studies, yet often lack appropriate statistical support. The authors are strongly encouraged to clearly enumerate what new information/insight their study provides that has not already been previously described (e.g. by Forslund et al, 10.1101/gr.155465.113)

Major comments:

1. Correlations between ARG burden and per capita antibiotic usage can be of great interest to the field and provide further support to previously identified similar patterns. However, the authors relied on reports from WHO and CDDEP as resources for antibiotic usage data, and they are not directly relevant to the cohort the authors analyzed. Also, consequently, the authors had to compare per capita antibiotic usage and “median” of the per sample ARG copies/genome. These factors may strongly mislead the reported correlation analyses.
2. What is the rationale for using nucleotide sequences of the ARG ORFs for clustering instead of amino acid sequences? Given codon biases across taxonomic levels, it is more appropriate to cluster ORFs using amino acid sequences
3. Similarly, ARG novelty that the authors demonstrated in the manuscript was conceptualized considering nucleotide identity. Defining novelty of an ARG based on nucleotide identity (as opposed to amino acid ID) may be overreaching.
4. The authors argued that ARG diversity varies across body sites without any statistical support, which is unacceptable. Also, this ARG diversity comparison seems not relevant to or coherent with other parts of this study.
5. Subsampling depths for rarefaction analysis (Fig. S2) are uneven. Repeat the analysis using identical subsampling levels.
6. The text reads to suggest that ORFs were identified in MAGs. As such, assigning ORFs to be of plasmidic origin (line 271) is inappropriate. If all assembled metagenomic contigs (including non-genomic contigs) were used to determine ORFs, please clarify the appropriate section to indicate that.
7. Strain pathogenicity is a highly complex trait and has high within-species variability depending on innumerable factors. As such, classification of SGBs as pathogenic or non-pathogenic depending on previous reports of infection at any site in human body is inappropriate. Claims regarding pathogenicity of SGBs need to be revised or further justification needs to be provided for the current classification.

8. For putative mobile ARGs, the authors should look for similarities in genome contexts within which such genes are located to provide further support of mobilization
9. Overall, I found the manuscript difficult to read due to numerous apparent grammatical errors or typos, and a careful review/edit is suggested.

Minor comments:

1. The authors need to add full names of the first abbreviations in the manuscript and each figure caption.
2. i.e. -> i.e., (line 143, 211, 266, and so on)
3. The sentence at line 155-156 requires a reference.
4. The run-on sentences (lines 144, 459, 501, 540) and typos (lines 96) need to be fixed.
5. In the x-axes labels of Fig. S2A,C, change "reads" to "bases."
6. Show the figure for the correlation of the antibiotic use data for the 12 countries from the two databases (line 215).
7. Are the differences in diversities reported in Fig. S2 statistically significant? Report statistical tests and results.
8. What is MGE (line 268)?
9. a -> the (line 268)
10. In order, to -> To (line 325)
11. phenomenon -> phenomena (line 517, 583)

Response to reviewers

Reviewer 1

In this manuscript, Lee et al investigate correlations between population based antibiotic usage and the proportion of ARGs in human gut microbiomes. Finding a positive correlation between these two, they then attempt to link ARGs and MAGs. One overall limitation of this study is the focus on human antibiotic usage. The anomalously high resistance levels in China, for example, might be the result of antibiotic use in other industries (such as the livestock industry).

We agree that the anomalously high-levels of ARGs in China could be due to antibiotic usage in the livestock industry and we have now expanded on this. in the Discussion (see Lines 610-612). However, we disagree that focusing on the better documented rates of human consumption is necessarily a major limitation of this study. We do not believe that comparable statistics on livestock antibiotic usage are readily available and moreover the strength of the overall correlation with human consumption implies that this is actually the major driver of resistance in the human microbiome, indeed that is what enables us to identify China as an outlier.

The authors then identify a major and minor resistotype in a nice cluster-based analysis. The minor resistotype is associated with a higher abundance of proteobacteria and more resistance gene types, as one would expect/predict. While the authors took good care to remove samples from subjects who may have been exposed to antibiotics at the time of sampling, I wonder if the minor resistotype reflect subjects who recently were exposed to antibiotics (and thus have an enrichment of proteobacteria that harbor more numerous and diverse ARGs).

We now term the minor resistotype the 'FAMP resistotype', since after re-analysis using a read-based ARG profiling approach (see below) it is somewhat higher frequency than in our original analysis (44.6% vs. 26%) and, hence, not unequivocally the minority resistotype. This FAMP resistotype is indeed associated with antibiotic consumption which induces a transient shift towards the FAMP state over a short time scale (Figure 6) and the FAMP associated ARGs are principally derived from Proteobacteria. We agree therefore that it could be a result of individuals that were recently exposed to antibiotics, that we cannot completely discount this is one of the limitations of this type of meta-analysis. However, there are number of pieces of evidence that convince us that the FAMP resistotype occurs even in individuals that have not recently consumed antibiotics. Firstly, across the whole data set 28.6% of control individuals derived from the FAMP resistotype, and crucially when we stratified the subjects by the documented antibiotic-free period that they had prior to sampling, 22% of the healthy subjects that had been recruited with the exclusion criterion of demanding at least 3 months of antibiotic-free period prior to the sampling were classified as the FAMP resistotype (Figure S11B). Secondly, if we are simply observing the after effects of antibiotic consumption by individuals then we would expect in the healthy controls that the proportion of the FAMP resistotype in a country would correlate with consumption rates. The fact that we do not see such a correlation convinces us that this is a genuine population level rather than individual response. Finally, whilst the FAMP associated ARGs derive from Proteobacteria, the FAMP resistotype itself is not primarily driven by community composition, only 1% of the variation in species profile is explained by resistotype. Our hypothesis is that it is other phenomena (such as disease) which cause the increase in Proteobacteria but that the ARG load of those Proteobacteria is then dependent on the overall resistance carriage in the population and that determines the shift to the FAMP resistotype. We add some comments on these important points to the Discussion (see Lines 622-629).

Overall, I found the manuscript interesting and well written. I am hard pressed to understand and make sense of the fact that the conclusions of this manuscript suggest only two resistotypes whereas the Ruppe findings suggest many more.

There are several possible reasons for the higher number of resistotypes found in Ruppe’s study. Firstly, Ruppe’s method of identifying ARGs is likely more sensitive since they used structural homology. This allowed them to collect a huge catalogue of putative resistance genes that share typically 30% or lower sequence identity with the known ARGs databases. In contrast, we focused on close sequence homologs of known resistance genes, using 80% identity/coverage cutoff. Hence, the “gut resistome” analyzed in our study is a conservatively defined set of genes that are close homologs of experimentally characterised ARGs, whilst Ruppe’s “gut resistome” is a significantly more inclusive set of genes. The landscape of inter-individual variation in such differently defined “resistomes” is likely very different. Secondly, Ruppe et al. used a Dirichlet multinomial mixture model, while we used the partitioning around medoids method to determine clusters. The application of Dirichlet multinomial mixtures to continuous variables such as the abundance of resistance genes is not appropriate because the multinomial sampling explicitly assumes discrete counts, rather than the continuous coverage depths obtained from metagenomics analysis. This difference in clustering algorithm may also explain the discrepancy between our studies (see Lines 630-635 of the Discussion).

I wonder to what extent the minor resistotype (which looks a lot like the composition of microbiomes that are exposed to antibiotics) drives the MDS results and suggests only 2 clusters. Put another way, if that minor resistotype is removed and clustering is done again, does one see more subtle differences or sub clustering within the major resistotypes? On balance, this is a small point, but it would add some intrigue to the manuscript.

We examined if the major (renamed background) resistotype can be further split into more than one distinct sub-cluster. The figure inserted below shows the average silhouette width obtained from partitioning around medoids using k values from 2 to 20 (note silhouette width cannot be defined for k = 1) applied to just those samples in the background resistotype. This gives a maximum at two but we still need to determine if two clusters are more supported than just a single cluster. This cannot be answered solely by average silhouette width analysis.

To address this, we inspected the total within-cluster sum of squares for k values from 1 to 20. This quantity will always decrease with increasing cluster number but at the most appropriate k a distinct flattening or ‘elbow’ should be observed. For the samples in the background resistotype (left) we do not see a distinct elbow suggesting just one cluster is present, whilst the original division of the two resistotypes created a clear elbow at k = 2 (right).

Additionally, visual inspection using NMDS ordination of the samples within the major resistotype indicates that the two subclusters forced by the clustering method (with $k = 2$) are poorly separated from each other (see below).

Hence, we concluded that it is not appropriate to see the background resistotype as a complex of two or more distinctive subclusters. This is also what we would expect if the original clustering was valid, if partitioning the background resistotype was appropriate we would have predicted more than two resistotypes to begin with but this is a good sanity check of our analysis.

As it stands, an antibiotic usage expert with a microbiology background would probably be unsurprised by the results presented here - that there is a resistotype that is enriched for ARGs and that the ARGs are largely proteobacteria-associated. I am thus wondering if there are ways to expand upon the very nice analysis performed here to provide a reader with more insight - thus increasing the impact of the manuscript. Below, I offer several major and minor comments on ways to potentially accomplish this objective.

Major:

1. One of the limitations of the approach used here, where ARGs are identified from the assemblies as opposed to the reads is that the ARG has to be in a sufficiently well covered genomic region/MGE to be assembled well (usually covered $>\sim 5x$). This limitation should be made clear in the text. It would be interesting to compare the assembly based results vs. short read based ARG quantification approaches. For example, the finding of a median of only 14 ARGs per metagenome seems low. How would this compare to a method that uses reads to call ARGs (as opposed to assemblies)? I anticipate the latter would be more sensitive.

This issue was raised by multiple reviewers so we have addressed it in detail. We directly compared for the 5,341 adult stool samples in our study for which raw reads were available from the NCBI or ENA, the ARG profiles predicted by mapping individual reads to the CARD with the original assembly-based approach both in terms of diversity and abundance. This required significant computation but as expected by the reviewers (see the newly added Figure S3A) we did indeed observe a greater diversity of ARG families from mapping reads in some samples. The read-based method also produced higher total ARG abundance (cpg) per sample, although the total abundance (cpg) from the two methods had a strong correlation (see Figure S3B). We then compared individual ARG abundances between the two methods (Figure S3C) observing 5,742 ARGs that were detected by read mapping that were missed by mapping to assemblies, this and the distribution of ARG family abundances (Figure S3D), confirmed that it is the low abundance families that were missed by the assembly-based approach.

Based on these observations we concluded that the read-based approach is indeed more sensitive at detecting low-abundance ARGs. Consequently, we repeated all analyses in our study based exclusively on ARG family abundances with the read-based calculation, these include the country-level variation in total ARG abundance among the stool metagenomes and its correlation with country-level antibiotic usage statistics (Figure 2, Table S4, Table S5), clustering of resistome profile variations into resistotypes (Figure 3, Figure S8), and the downstream

analyses that rely on the resistotype assignment (Figure 4, Figure 5, Figure 6, Figure 7, Figure S9, Figure S10, Figure S11, Table S7, Table S9).

None of the key messages i.e., the presence of two resistotypes and the country-level correlation between ARG abundance and antibiotic usage, changed after applying the read-based abundance profiles. One significant change regarding this methodological adjustment is that the population level proportion of the two resistotypes became less pronounced than previously observed (74% vs. 26% based on previous assembly-based; 55.4% vs. 44.6% based on the updated read-based profiles). The characteristic features of the secondary (FAMP) resistotype did not change – still the same ARG families, ARG classes, and species are enriched and there remains a gradient of the FAMP frequency from healthy to disease. We removed the “major” vs. “minor” distinction from the naming of the two resistotypes and renamed them as the “background” and the “FAMP” resistotypes.

Not all the results from our assembly-based approach can be replaced with the read-based analyses. The generation of the clustered catalogue of ARGs, estimates of ARG cluster diversity, that depend on the associations the ARGs have with species-level genome bins, taxonomy, and/or mobile genetic elements can only be obtained by an assembly-based approach. Those results were kept unchanged.

2. The authors correctly point out that methods like HiC are required to map plasmid MGEs to chromosomes; alternatively, methods like long read sequencing can incorporate integrated MGEs into chromosomal DNA. Both plasmids and integrated MGEs are underrepresented in binned MAGs; thus the approach outlined to investigate MAGs likely undercalls the proportion of ARGs that are associated with genomes. What proportion of the ARGs, for example, are found in the “unbinned” fraction?

It is true that MGEs and the ARGs therein may be underrepresented in MAGs. To address the reviewer’s question, we reorganized Table S6 to include the proportions of unbinned ORFs in each ARG category. Note that we considered an ORF to be binned only when the ORF was contained in a high-quality MAG. Overall, 64.5% of the ARG ORFs annotated in the adult stool metagenomes were unbinned, and the unbinned proportion was relatively low among fluoroquinolone- (34.4%), peptide antibiotics- (37.8%), and multi-drug-resistance genes (37.9%), while relatively high among sulfonamide- (93.2%) and amphenicol-resistance genes (91.9%). This trend is in agreement with our analysis of the proportion of mobilized (multi-species) ARG_cluster99 in ARG families, where fluoroquinolone resistance gene families contain the lowest fraction of mobilized clusters and the sulfonamide resistance families the highest (Figure S5A).

The unbinned ARGs may obscure the host range of mobile ARGs across taxa. We supplemented the mapping of ARG clusters (particularly ARG_cluster99s) to the host taxonomy by incorporating >150,000 RefSeq genomes generated from isolates. A key point though is that to determine the host range of an ARG cluster, we only need one ORF in each cluster to be binned into a MAG of that species (SGB), even if binning fails for an individual ARG a majority of the time, it does not matter to these statistics. This is confirmed by two observations firstly that there was a very good correspondence for individual ARG families between the proportion of RefSeq ORFs assigned to multi-species clusters and the ORFs from adult gut microbiomes (Figure S5E). Since, RefSeq ORFs will be linked to correct genome this implies that the binning failures are not dramatically distorting this statistic. Secondly, 89.1% of plasmid borne ORFs were assigned to multi-species clusters, since there are some species-specific plasmids this places an upper limit on the false negative rate of multi-species cluster detection of around 10%.

3. The authors do attempt to identify plasmid-associated ARGs; but do not have a specific approach to account for integrated HGT segments that can integrate into genomes and carry ARGs. This is an important source of an error by omission.

To address this omission, we detected hallmark genes associated with class 1 integrons, insertion sequence (IS) elements and prokaryotic conjugation systems on the assembled metagenomic contigs and RefSeq genomes on which we have analyzed ARGs. Based on the distances to the signature genes of these mobile genetic elements, we classified each ARG ORF into either “associated” or “not associated” with the given MGE class. We found that 36.2% of the adult gut resistome ORFs were associated with at least one type of MGE (27.1% on plasmids; 4.4% proximate to IS transposase; 10.2% proximate to conjugative systems; 0.3% proximate to integron integrase) and 39.2% of the RefSeq genome ARGs were associated with at least one of the MGEs. The results have been now integrated into the revised manuscript as we previously observed for plasmids there is a strong correlation between nearby MGE signatures and assignment of an ARG to multi-species clusters. (Figure S6). We were also able to quantify the relative importance of these different MGEs in mobilizing ARGs (Lines 286-312).

4. Is the ARG abundance perhaps related to issues beyond antibiotic usage per capita; For example, the amount of antibiotics used in rearing of livestock, the prevalence and range of antibiotics found in drinking water, etc? While the positive correlation observed is interesting, it is likely that many features beyond antibiotic usage per capita are contributing to this finding.

We agree that other factors will impact the abundance and diversity of ARGs at a country-level. However, there is a limit to what can be done in a single study and we believe that the strong correlations observed here demonstrate that human antibiotic consumption is the most important single factor.

5. The rarefaction that was performed to account for differences in sample number across body sites was a good idea; one thing that isn’t well accounted for is the known variation in skin microbiome composition at different skin sites. For example, the palm of the hand has different microbes from the retroauricular fold. Thus, if all of the skin samples (or most) were from one body site, there still may be undiscovered diversity of ARGs in other skin sites on the body.

Differences in the skin microbiome across body sites is something we could have paid more attention to. To address the concern that the aggregate skin resistome diversity represented in the current results might not be representative of the overall skin resistome if the samples have bias to certain locations on the human body, we first summarized the distribution of the skin samples analyzed in our study across the human body. The samples were evenly distributed across various locations rather than skewed to a specific location (see the figure below).

As the reviewer anticipated, the skin resistome did vary depending on the location of sample.

Comparison		Adonis2	
Body site 1	Body site 2	R2	FDR
Head face	Head other	0.058	0.0058
Head face	Lower limbs	0.084	0.0014
Head face	Shoulder	0.11	0.0014
Head face	Torso	0.047	0.012
Head face	Upper limbs	0.056	0.0025
Head other	Lower limbs	0.10	0.0014
Head other	Shoulder	0.20	0.0014
Head other	Torso	0.099	0.0014
Head other	Upper limbs	0.15	0.0014
Lower limbs	Shoulder	0.10	0.0014
Lower limbs	Torso	0.054	0.0014
Lower limbs	Upper limbs	0.085	0.0014
Shoulder	Torso	0.065	0.0014
Shoulder	Upper limbs	0.082	0.0014
Torso	Upper limbs	0.025	0.054

Even the rarefied diversity was substantially different between the different sample locations. Nonetheless, the results also confirm that the diversity estimated based on the aggregate of all skin samples reflected the average values across whole body sites. In our opinion, our approach of an aggregated diversity of skin resistome is the optimal choice when the purpose is to compare different body sites at a higher level (i.e., gut versus skin versus oral, etc.).

A finer scale analysis across sites would be interesting but would add to an already long paper and there are not that many samples available for each body site.

6. I am curious to know whether the ARGs that were mappable to taxa were different classes than those that were not mappable to taxa (line 257). For example, I would predict that tetR genes often were mapped to taxa whereas beta lactamases were often not mappable to taxa (as they are more likely to be on MGEs). This analysis might help define which ARGs are more likely to be found on MGEs and thus are also more likely to be transferred between species.

There is substantial variation in both the binning rate and the signals of mobilization (i.e., plasmid localization and multi-species cluster) across ARG families, and these are often clearly polarized into mobile and non-mobile ends of the spectrum. We have summarized detailed characterization of ARG families regarding binning rate and signatures of mobilizations in the new Supplementary File 6. Regarding the specific gene that the reviewer mentioned, tetR was not included in our ARG annotations as it is a “protein overexpression model” rather than “protein homology model” in the CARD. Some beta-lactamases do indeed have low mapping rates to taxa (low binning rate), for example, the OXA-347 gene family ORFs in the gut metagenomes (n = 956) had 4.3% binning rate. There are also beta-lactamases that tend to be mappable to taxa with higher binning rate, such as cblA (n = 4,011 ORFs; binned in 59% of the cases).

7. Figure 7 - it is interesting to note that the central module contains genes that are frequently shared between organisms in the central module, but that these are almost never shared with pathogens. This suggests that gene transfer is happening between related and unrelated gut microbes (commensal type) but that this isn't necessarily leading to an enrichment of ARGs in pathogens (which seems to be the premise for the study).

There are some pathogens in the central module (9% vs 24% in the periphery) but certainly the non-pathogens are enriched as are multi-species clusters. So we agree that the impact of antibiotic consumption on ARGs in

pathogens may be less than anticipated, this is an important conclusion and we have now emphasized this (Lines 657-664).

8. I am not an expert statistical reviewer, but the analyses performed seem sound and generally well grounded. That said, it would be valuable to seek comments of an expert statistical reviewer.

We have done our best to use appropriate statistical methods throughout.

9. There are many statements of novelty in the manuscript (esp in the intro and discussion); given this, it would be helpful for the reader to see a supplemental table that provides an overview of the studies of ARGs in the human microbiome to date. Alternatively, if this journal does not allow statements of novelty, perhaps these comments should be removed.

We removed the phrase ‘this is the first time to our knowledge ...’ from the Discussion (Line 604) and the novelty statement on the size of the study see Abstract and Line 568, these statements are still true to the best of our knowledge but we hope this satisfies the reviewer.

Minor

1. ‘Form’ should be ‘from - line 96
2. Microbiome is misspelled on line 96
3. Given that only 1 milk sample is used, it would seem reasonable to omit this sample type; or to expand to a larger resource/database that has more milk samples.

We made these corrections and removed the milk sample.

4. How was the 80% identity cutoff / 80% target sequence length (line 104) selected?

Though any choice will have a degree of arbitrariness 80% identity cutoff is often considered a good stringent threshold in homology-based identification of antibiotic resistance genes (10.1186/s40168-018-0401-z). Furthermore, we curated the CARD database reference proteins into ‘ARG family’ units, which we defined based on 80%-identity clustering combined with manual inspection of the phylogenetic tree and the annotation given by CARD. Therefore, it seemed logical to also use an 80% identity cut-off for annotation of novel genes and assignment of that ‘ARG family’ to a new ORF or aggregate query reads. The length breadth-of-coverage cutoff of 80% is also a commonly used threshold value (UniProt uses 80% length coverage in clustering of UniRef, for example) to ensure a functionally meaningful sequence homology.

5. I don’t know that we are certain that the gut is the most “metabolically active” part of the microbiome; I might recommend deleting this phrase (line 155).

This has been deleted.

6. One limitation of excluding infants and children from this analysis is that children are often exposed to antibiotics; and thus might be more likely to harbor ARGs that may be lost by adulthood. I wonder if it might be worth redoing this analysis to include infants/children; or to do an additional separate analysis with this cohort.

We are aware of the importance of understanding the development of resistomes in early life, especially given the frequent exposure of infants to antibiotics. We decided to exclude infants and children from the analysis of resistotype clustering or country-level correlations for two reasons, firstly, a lack of sufficient sample numbers for many countries, and secondly, a higher level of variability compared to the adult resistomes. The table below demonstrates that the newborns and children in this data set, are rather sparsely distributed across countries.

Continent	Country	Excluded age groups		Included age groups ("adult" collectively)		
		Newborn 0 ≤ age years < 1	Child 1 ≤ age years < 12	School age 12 ≤ age years < 19	Adult 19 ≤ age years < 66	Senior 66 ≤ age years
Africa	MDG			1	110	1
	TZA		7	6	48	1
N. America	CAN				119	193
	USA	37	3	6	404	55
S. America	PER	1	16	2	16	
Asia	BGD		13		30	
	CHN			8	1,107	181
	KAZ				168	
Asia/Europe	MNG				110	
	ISR				956	
	RUS	269	29			
Europe	AUT				60	94
	DEU		23	3	243	22
	DNK				387	16
	ESP			5	345	6
	EST	177	42			1
	FIN	251	126			1
	FRA				91	66
	GBR				178	72
	HUN					1
	ISL					1
	ITA	103			42	
	NLD			60	448	12
	NOR					1
SVK					3	
SWE	200	99		169	130	
Oceania	FJI	5	34	18	112	3

Secondly, the figure below shows that the total ARG abundance and the diversity of ARG_cluster99s are both significantly different amongst the age groups (Kruskal-Wallis test; $P = 3.6E-23$ for abundance; $P = 3.8E-87$ for diversity) and especially in the case of diversity, the newborns exhibit more variation than the adults. The composition of ARG families also varied across the age groups (Adonis2; $R^2 = 0.056$, $P = 0.001$) and the newborns and the children tend to appear as extreme outliers on the NMDS plot.

7. Line 167 onward - While the healthy controls may not have a diagnosis of a given disease that was studied in the parent study, it is important to note that when aggregating data from other studies, “healthy” is not a term that is uniformly defined across studies. For example, in a cancer study, a subject who has hypertension and hyperlipidemia, but has never had cancer, may be counted as a “healthy control”.

This is a good point, we, as noted in the original methods, did for a number of studies ensure that controls were indeed healthy but it could remain an issue, and we have noted this (Line 924-926). We have also in several places replaced ‘healthy’ with ‘healthy control’ to better emphasise this.

8. I am curious about the distribution of the ARG copies per genome - is it unimodal, bimodal, normal or skewed? It would be nice to see these data in a figure or supplemental figure. Some resistance genes (e.g. tetR genes) are often chromosomal and would be expected to be in a single or few copies per genome range. Other ARGs (on mobile elements, etc) might be present in varied copy number (potentially even very high cpg).

It is important to note that the copies per genome (cpg) values we calculated to quantify the relative abundance of ARGs tell us the community level average but does not partition that across different genomes. We appreciate the reviewer’s curiosity and agree with the reviewer’s expectations that regulatory, chromosomal resistance genes would tend to have low copy numbers while mobile ARGs would sometimes have very high copy numbers. To properly address this, we would have to analyze the variation of depth within/among the contigs in the MAGs (or RefSeq genomes). Since in most samples most ARGs are not in MAGs this is difficult.

9. One thought is that ARGs/metagenome may be higher in more diverse samples or individuals that have a long tail distribution of strains/species. How is this accounted for in the analysis? For example, Individuals in France

might have a high number of observed ARG clusters but that may be due to having more diverse taxa (at a more even representation and thus more likely to be assembled into contigs). This is why a read based approach may also be helpful in teasing these relationships apart. I believe that the results presented are likely correct - but these are confounders that should be considered and would be helpful for the reader to see and understand. The cpg analysis somewhat accounts for this, but assumes that all organisms (or at least most of them) from a given sample have a MAG. This is almost never the case given the standard range of sequencing depth that is typically pursued.

We have checked that the country-level ARG diversity trends were not driven by the difference in species diversity. Firstly, there was no correlation between the species richness (unrarefied number of observed species) in the metagenomes from the countries and the ARG_cluster99 richness (Spearman's rho = -0.073, P = 0.84; Kendall's tau = -0.018, P = 1) (see the figure inserted below).

The absence of correlation between the species richness and ARG_cluster99 richness was still true after re-calculating both values simulating a reduced sequencing depth for all samples of 1 Gbp, removing the ORFs (either SCG or ARG) detected on contigs whose depth went below 1.5X (our estimated limit of detection) after subsampling. In the resulting rarefied species richness and ARG_cluster99 richness data (see the figure inserted below), the correlation between median species richness and median ARG_cluster99 richness across the countries was insignificant (Spearman's rho = -0.18, P = 0.60; Kendall's tau = -0.18, P = 0.43) but still the correlation between median ARG_cluster99 richness and the country level antibiotic usage (WHO statistics) was as strong as we originally reported in the manuscript (Spearman's rho = 0.83, P = 0.015; Kendall's tau = 0.71, P = 0.014).

We therefore conclude that difference in species diversity are not driving these phenomena.

10. Figure 3 - It would be helpful if FAMP was defined in the legend (for readers who will simply scan through the figures and legends).

We added the following line as a definition of FAMP in the legend for Figure 3D: “Naming of the FAMP resistotype reflects the five antibiotic resistance classes enriched with the largest fold difference: F, fluoroquinolone and fosfomycin; A, aminoglycoside; M, multi-drug; P, peptide.”

11. Figure 4 - perhaps I missed this, but what resource was used to categorize “pathogens” vs. “non-pathogens”? I don’t have a particular problem with this designation, but as this is a hot topic issue (using the term pathogen as a binary category) in the microbiology community, it would be helpful to make clear how this categorization is being done. As an example, *Bacteroides fragilis* can be a pathogen, but typically is not and is a common member of gut microbiomes. So would it be categorized as a pathogen or a non-pathogen for the purpose of this analysis?

This question was also raised Reviewer 2 so we address it in more detail there. We wanted a definition of pathogen that included any microbe that may be the target of antibiotics at the species level. The point being that recombination of ARGs will occur rapidly within species, hence, in order to determine what drives antibiotic response that seemed most appropriate. Therefore, we generated a list of bacterial species with reported causal role in human infections from the 11th edition of the ASM’s manual of clinical microbiology. *Bacteroides fragilis* was categorized as a pathogen according to this definition. We have added a discussion of this point (Lines 429-434).

12. What were the cutoffs for sequencing depth (bp sequenced) for inclusion in this study? Also, what was the range of coverage? The results are sensitive to variations in this. If one country, for example, had samples that were less deeply sequenced than another, one might erroneously conclude that there are fewer ARGs in those metagenomes.

We filtered out all samples where we failed to detect all 40 prokaryotic single-copy core genes on the basis that these were failed sequencing runs. Regarding the abundance of ARGs, different sequencing depths are addressed by normalizing with respect to the median SCG coverage depth to obtain copies per genome (cpg). To account for different sequencing depths when addressing diversity of ARGs, we performed random sampling of samples so that when comparing across countries, the total depth sequenced was the same (Lines 840 – 864).

13. I appreciate that PlasmidNet is a modification of the PhageBoost algorithm; to be included here, it would be helpful if PlasmidNet was benchmarked properly and described in terms of its performance characteristics. Else it is hard to know how to evaluate results from this tool.

Benchmarking results for PlasmidNet have now been added to the manuscript (Lines 877-879).

Reviewer 2

The article by Lee et al., analyzed human metagenomes in order to understand how antibiotic resistance genes (ARGs) are distributed among different human cohorts and countries, and what factors drive the patterns such as the usage of antibiotics at the country level. While several of the findings reported such as correlation with antibiotic usage, frequent horizontal transfer of ARGs, duration of the effects of the antibiotic administration on the microbiome, echoed previous findings and thus do not represent novel results, the scale that this study was performed and the results provided in terms of quantification represent novel contributions, to the knowledge of this reviewer. Further, the two types of resistomes reported also represent interesting findings. The paper is overall well written and does not require much editorial editing albeit it is quite long (I made some suggestions about shortening below). That said, there are several methodological and interpretation issues that require more attention by the authors. Most notably, the approach for estimating ARG abundance, the lack of correlation between a specific ARG and the usage of its corresponding antibiotic, and the possible effect of diet or asymptomatic infections on the resistome types and the outlier Chinese samples (see also specific points below). Further, is the minor resistome driven mostly by Chinese samples? Is this likely due to diet e.g., higher frequency of hot spicy food (some spices are related to antibiotics, I think)?

We appreciate the overall positive comments from the reviewer and we address their specific issues regarding ARG annotation, China as an outlier, antibiotic class level correlations, etc. below. The FAMP resistotype is higher in the Chinese samples and a possible association with diet is an intriguing hypothesis to explain this but given the difficulty quantifying that, in the absence of dietary meta data, we think it is better left to a follow-up study.

Minor/Specific points

“Proteobacteria” in the abstract and other taxonomic names elsewhere should be in italics.

We have now italicized taxonomic names throughout the manuscript.

Lines 63-67. It seems that the justification for the study is not very strong, as stated here. Perhaps the authors could highlight more that the quantitative aspect is lacking in previous publications, and the usefulness of different “states” or “types” of resistomes for antibiotic usage and management. I do think that the study has enough novelty although this is probably undersold here.

Line85-90. These do not represent news really. The authors could focus more on the novel findings reported in their manuscript, which could help reducing the length of the paper and thus, make it easier to read.

In our opinion these results are novel, we can’t find any earlier publication that demonstrates these correlations, the association with a central component of the taxa-gene network or the two resistotypes. We are happy to revise if we have missed some of the literature?

Line 113. What "unique" ARGs means here? It is not at 95% nt identity as the previous sentence indicate, right? Also, the previous sentence needs a citation to back up the threshold used.

'Unique' ARG sequences here means non-redundant at the sequence level, any differences in nucleotides generates a new ARG determined by clustering with 100% DNA sequence identity cutoff. We added a citation (10.1038/s41586-021-04233-4) to the preceding sentence.

Lines 142-146. Interesting data and I think the paper has enough novel results and thus, could make a stronger justification around quantification. (Related to the major suggestion above)

Thanks, we have tried to emphasis this better in the introduction.

Lines 180-185. Is it possible that the higher diversity signal in the Chinese samples is due to more people sampled or in more remote/not connected areas? Or diet and spicy foods?

It is not possible for us to evaluate or rule out the possibility that dietary factors or socio/geographic factors at intra-country level might have driven the resistome diversity in certain countries, due to the lack of metadata. We did test for the other possible confounding factors, in light of this comment. Firstly, sample size and general depth of sequencing. The number of healthy stool metagenomes from the healthy adults not taking antibiotics – which are what matters here because the ARG diversity was compared amongst those subjects – from China was 340. This was not particularly a large number, it is exceeded by other countries (956 from Israel, 470 from the Netherlands, 401 from Denmark) (see Table S3). The sequencing depth of each individual stool metagenome from China was also not particularly deep compared to the metagenomes from the other countries, in fact, the sequencing depth of the Chinese samples (median 3.7 Gbp; inter-quartile 2.7 – 4.9 Gbp) was on the lower side of the spectrum (see the figure below). We should also note, as above, that our subsampling procedure will correct for different average sequencing depths between countries.

Secondly, if higher geographic diversity in China had inflated the observed ARG diversity, we would expect a similar effect to be seen in species compositions, i.e. Chinese samples would display greater inter-individual variation than the samples within other countries. In terms of Bray-Curtis dissimilarities among the samples from each country, China was ranked 5th out of 11 countries included in the country-level ARG_cluster99 diversity comparison, hence no evidence that the Chinese samples were particularly heterogeneous (see the below figure).

Line 190-193. There is potentially an issue with how abundance is calculated, and this could possibly underly the lack of signal/correlation between the abundance of specific ARG and the usage of their corresponding antibiotic. That is, to take the abundance of the contig as the abundance of the ARG is potentially problematic because it depends on the coverage (and the nature) of each gene on the contig (i.e., better to estimate abundance on a per ARG gene basis). Further, it would be better to map the reads on ARG sequences with a threshold for identity, and not rely on the assembly because the assembly identity threshold is not defined and could be variable, and lower than the 99% threshold used to defined ARG variants. Calling ARG on contigs seems OK/robust. In the view of this reviewer, it would be helpful for the authors to validate their abundance estimation method and show that any biases did not significantly affect their results and conclusions. This does not have to take place for all genes, which is computationally very expensive, but just for a small subset of genes and samples/metagenomes.

We address this comment in our responses to Reviewer 1 above, in summary, we did this comparison (see Figure S3) and decided the reviewers were correct, direct read mapping to the CARD was more sensitive. Therefore, for all the ARG abundance statistics we recalculated everything to use read-based results. In fact, having done this at the class level we did see a significant correlation between beta-lactam abundance and consumption as the reviewer suspected.

Line 257-233. Or that the abundance estimates are not highly reliable at the individual ARG family? See related comment just above. This result reported here does not make much sense and the authors should investigate it further, I believe.

See comment we do now see a correlation for beta-lactams but not for the other classes, beta-lactams are the predominant clinical antibiotic class so this makes sense.

Line 325. No comma needed at “In order, to”

Line 407 and elsewhere. I would not call them pathogens by the taxonomy only, especially for enterics; e.g., I would look, in addition, for presence of the known virulence genes. Consider the E. coli case for instance; some E. coli are commensal and do not carry the major virulence factors such as toxins, hemolysins etc.

As we discuss in response to Reviewer 1 above, we wanted an inclusive definition of a pathogen that at the species-level i.e. SGBs included all taxa where any strain may be a pathogen under any circumstances. In order to identify species that are the potential clinical targets of antibiotics. We agree that identifying virulence factors directly on genomes might be more precise but at the species levels there is a good correspondence with our current method. We demonstrate this below by identifying virulence genes in RefSeq genomes or MAGs and then

for each SGB giving the distribution of the number of virulence factors (top total number, bottom those directly implicated in disease) observed and then colouring SGBs according to whether they were pathogenic under our definition. The majority of SGBs that were define as pathogens have high numbers of virulence factors and vice versa. Therefore, switching to the use of virulence at the species level would not greatly change our results in a statistical sense.

Related to the pathogen work: is it possible that the minor resistome represents bacterial infections, and not healthy states as originally thought? Please double check the metadata for the corresponding samples.

As we discuss, Lines 490 – 505, the FAMP resistotype is more prevalent in individuals with enteric infections. However, we do not believe that all individuals with the FAMP resistotype have bacterial infections in a clinical sense. Firstly, because as we discuss in the text, it is prevalent (28.6%) in the ‘healthy control’ groups, of course they may have undiagnosed and undocumented infections, we cannot account for that, but it seems unlikely at that frequency. Furthermore, we know that the subjects in the short term antibiotic treatment experiments were healthy and antibiotics caused a shift to the FAMP resistotype without them being clinically diagnosed as experiencing an enteric infection. Finally, if the FAMP resistotype was purely the result of infection we would expect a much

larger impact on community structure than we observe, only 1% of variation in microbiome is explained by resistotype – (Line 412). Our hypothesis is that FAMP resistotype reflects the presence of diverse Proteobacteria that have acquired a set of species-specific but clinically relevant resistance, that could be in individuals that have enteric infections, and those pathogenic gram-negatives are more likely to be resistant and hence shift the individual to the FAMP resistotype, but not in all cases. It could also be that commensal Proteobacteria have acquired these genes due to spill-over in otherwise healthy microbiomes. We have added some Discussion of this to the MS.

Lines 465-475. I suspect that the authors just repeated the results of the previous study here and thus, this section could be removed (if my suspicion is true).

In our analysis of the short-term impact of antibiotics on resistotype, we reanalyzed the published data set using our own ARG annotation strategy and the novel categorization of samples to resistotype. We cannot simply refer to their original results since they did not perform such an analysis.

Network analysis. I am not sure what the authors can infer from this network work; e.g., they can not know which node/population gives the ARG to which. Further, the composition of the network seems like the expected one based on the relative abundances of high vs minor resistome types and the membership of the types. Accordingly, this section could be reduced without much loss in clarity, and the discussion on line and on is more like hand-waving (since direction of horizontal gene transfer can not be established).

We disagree with the reviewer's assessment of the network analysis. It is true that directionality cannot be inferred but large-scale patterns in the degree of connectedness can, and it very clearly illustrates that there is a large connected component of commensals associated with the antibiotic consumption response, that is only weakly connected with the peripheral pathogens driving the FAMP resistotype. This separation of the two phenomenon is one of the central messages of the paper. We have tried to emphasise this more in the discussion though.

Line 532. phenomena, not -non.

Changed.

Line 540-542. There might be a diet effect as well e.g. more spicy food?

See response above, given the lack of dietary meta-data and the variability of diets within a country we cannot really answer that at this point.

Reviewer 3

In this manuscript, the authors conducted a large survey of antibiotic resistance genes (ARGs) in publicly available metagenome assemblies spanning multiple countries. The authors claim that the ARG richness in metagenomes positively correlates with the reported antibiotic usage rates in the countries under investigation. The authors claim to have detected networks of mobile ARGs that are transmitted among pathogenic and commensal members of the human microbiome. Furthermore, the resistomes are suggested to group into two distinct clusters: major and minor resistotypes. The minor resistotype is claimed to have higher ARG burden, consist of higher proportion of mobile ARGs, and be overrepresented in pathogenic organisms.

While this work analyses the largest dataset of this type, the conclusions often appear confirmatory of previous studies, yet often lack appropriate statistical support. The authors are strongly encouraged to clearly enumerate what new information/insight their study provides that has not already been previously described (e.g. by Forslund et al, 10.1101/gr.155465.113)

There are multiple insights that we provide beyond the Forslund et al. (2013) paper referenced above and their follow up review (Forslund et al. 2014 'Metagenomic insights into the human gut resistome and the forces that shape it'). Firstly, in these studies with the data sets available at that point, they were not able to demonstrate a significant association between total human antibiotic consumption and the overall abundance of ARGs. That may be because the sequencing was not deep enough or because they included both healthy controls and individuals with diabetes. They did observe a correlation with veterinary usage but they did not account for multiple comparisons and it is possible therefore that was artifactual result. They also included no analysis of ARG gene richness and no analysis of the transmission of ARGs between taxa, impossible to do in a comprehensive way without the MAG collection we exploited. That enabled us to demonstrate that the correlation we see with human antibiotic consumption is due to mobile multi-species ARGs. They also performed no cluster analysis of resistance profiles and hence did not find the two resistotypes we identify or link them again to the type of gene involved. We have tried to emphasis better in the introduction the novelty of our study.

We are perplexed by the statement that our conclusions lack statistical support when we extensively quote the results of statistical analyses and the methods used to achieve them. Out of the many results in our MS the reviewer identified one point where we omitted the test statistics, the body site comparisons, that was done simply to improve readability and is now rectified. In general, we include far more statistical basis for our conclusions than the papers cited by the reviewer. It is true that some of our results are confirmatory of previous studies but not all are, (see comment on the association with human consumption above) and in fact it would be disturbing if we did not see some of the same patterns as other studies. There is value in reconfirming an earlier result with more data and rigour.

Major comments:

1. Correlations between ARG burden and per capita antibiotic usage can be of great interest to the field and provide further support to previously identified similar patterns. However, the authors relied on reports from WHO and CDDEP as resources for antibiotic usage data, and they are not directly relevant to the cohort the authors analyzed. Also, consequently, the authors had to compare per capita antibiotic usage and “median” of the per sample ARG copies/genome. These factors may strongly mislead the reported correlation analyses.

The type of study the reviewer suggesting would be very impressive, to sample thousands of individuals all over the world, and combine that with an accurate long-term record of each person's antibiotic usage, diet, life-style and other contributing factors. Such a study though would be a massive logistical undertaking, beyond the scope of the resources of most research groups. Instead, what we are trying to do is add value to the existing metagenomics data sets that are publicly available. As to whether our method is valid, we believe it is for the

question we are trying to answer. We are not trying to explain individual variation in resistomes, because we lack the detailed meta-data, we are trying to explain the differences we observe between countries. The use of the population median to summarise the ARG abundance in a country in that context is valid. We agree that if our cohorts are not representative of the country then that will distort the results, but the fact that we do see strong correlations suggests that they are. We have though added this point to the Discussion as a possible explanation for China as an outlier.

2. What is the rationale for using nucleotide sequences of the ARG ORFs for clustering instead of amino acid sequences? Given codon biases across taxonomic levels, it is more appropriate to cluster ORFs using amino acid sequences

We compared both methods for ARG ORF clustering during the development of this study and we found better results from the use of nucleotide sequences rather than amino-acids. This is because we are using the clustering to detect recent horizontal gene transfer, our definition of a multi-species ARG is a 99% ARG cluster at the nucleotide level shared between two species (SGBs) defined at 95% nucleotide identity. The reviewer is correct codon bias will over time introduce nucleotide changes but it is actually exactly these changes that we wish to detect so that we can distinguish recent transfer from more ancient gene sharing. This approach was based on previous studies that rely on nucleotide sequence comparisons (Brito et al. 10.1038/nature18927; Groussin et al. 10.1016/j.cell.2021.02.052). In fact, amino acid sequences, because they experience stronger selective pressures may be less appropriate for this purpose. We want something as close as possible to a molecular clock so we can distinguish the divergence time of the core species genes from horizontally transmitted elements.

3. Similarly, ARG novelty that the authors demonstrated in the manuscript was conceptualized considering nucleotide identity. Defining novelty of an ARG based on nucleotide identity (as opposed to amino acid ID) may be overreaching.

This concern, that using nucleotide variation may overestimate functional novelty, was in fact our motivation for testing amino acid clustering as an alternative strategy. However, we found this not to be the case. When the same 99% identity – 90% alignment coverage thresholds were used to create ARG_cluster99s based on amino acid sequences rather than nucleotides, the proportion of novel clusters in the microbiome ARG catalogue was actually increased to 67.9%, compared to the value of 60.9% from nucleotide-based clustering in Table S2. We also observed similar rarefaction curves at lower identity cut-offs with amino acid clustering indicating that the ARG diversity was not saturating.

4. The authors argued that ARG diversity varies across body sites without any statistical support, which is unacceptable. Also, this ARG diversity comparison seems not relevant to or coherent with other parts of this study.

We added statistical support in Lines 150-153 of the revised manuscript. We disagree, although it is not our main focus it is useful to place the results on the gut resistome in the context of other body sites and in fact Reviewer 2 thought we should emphasise these results more in the introduction.

5. Subsampling depths for rarefaction analysis (Fig. S2) are uneven. Repeat the analysis using identical subsampling levels.

We have recreated all the subpanels in Figure S2 applying even increments of 10 samples.

6. The text reads to suggest that ORFs were identified in MAGs. As such, assigning ORFs to be of plasmidic origin (line 271) is inappropriate. If all assembled metagenomic contigs (including non-genomic contigs) were used to determine ORFs, please clarify the appropriate section to indicate that.

We determined ORFs on all metagenomic contigs regardless of whether that contig was binned into a MAG or not. We state that clearly in the text of the main paper - ‘We created a comprehensive catalogue of ARGs across both the human microbiome and reference genomes by locating open reading frames (ORFs) on metagenomic assemblies’ - and in the Methods sections and for convenience the workflow is summarised in Figure 1.

7. Strain pathogenicity is a highly complex trait and has high within-species variability depending on innumerable factors. As such, classification of SGBs as pathogenic or non-pathogenic depending on previous reports of infection at any site in human body is inappropriate. Claims regarding pathogenicity of SGBs need to be revised or further justification needs to be provided for the current classification.

We address this point in detail in the response to Reviewer 2, we show that there is a good correspondence between our definition of pathogenicity and the identification of virulence factors in the genomes or MAGs.

8. For putative mobile ARGs, the authors should look for similarities in genome contexts within which such genes are located to provide further support of mobilisation

We have added sequence based classification of transposons, conjugative elements and integrons, these and the original plasmid classifications are consistent with our multi-species definition of mobility in that the majority of ARG ORFS near such elements are mobile.

9. Overall, I found the manuscript difficult to read due to numerous apparent grammatical errors or typos, and a careful review/edit is suggested.

We have edited the MS extensively.

Minor comments:

1. The authors need to add full names of the first abbreviations in the manuscript and each figure caption.

This has been done.

2. i.e. -> i.e., (line 143, 211, 266, and so on)

A comma after i.e. is optional in British English.

3. The sentence at line 155-156 requires a reference.

Removed claim that the gut microbiome is the most metabolically active in response to similar comment above.

4. The run-on sentences (lines 144, 459, 501, 540) and typos (lines 96) need to be fixed.

These sentences have been shortened.

5. In the x-axes labels of Fig. S2A,C, change “reads” to “bases.”

Modified as suggested.

6. Show the figure for the correlation of the antibiotic use data for the 12 countries from the two databases (line 215).

Added the figure as a new Figure S4B.

7. Are the differences in diversities reported in Fig. S2 statistically significant? Report statistical tests and results.

Statistical test results were added to the manuscript.

8. What is MGE (line 268)?

Meant plasmid, specifically. Now the paragraph containing this sentence has been updated.

9. a -> the (line 268)

10. In order, to -> To (line 325)

Corrected.

11. phenomenon -> phenomena (line 517, 583)

Corrected.

Important: In addition to the above, you must comply with the following editorial requests; we will not be able to proceed with your revised manuscript otherwise. Please also see the Nature Communications formatting instructions, which you may find useful while preparing your revised manuscript.

POLICIES AND FORMS REQUIRED FOR RESUBMISSION

* Please complete or update the following checklist(s) to verify compliance with our research ethics and data reporting standards. Address all points on the checklist, revising your manuscript in response to the points if needed. The form(s) must be downloaded and completed in Adobe Reader rather than opened in a web browser. Each form must be uploaded as a Related Manuscript file at the time of resubmission.

[Editorial policy checklist](https://www.nature.com/documents/nr-editorial-policy-checklist.pdf): <https://www.nature.com/documents/nr-editorial-policy-checklist.pdf>

[Reporting summary](https://www.nature.com/documents/nr-reporting-summary.pdf): <https://www.nature.com/documents/nr-reporting-summary.pdf>

* Your paper uses custom code/software. Please complete the following code and software submission checklist and make your code available for reviewer assessment, if you have not already done so. The code/software can be provided in a zip file with a readme.txt file or other instructions for installing and running the software. If appropriate, also provide example data and expected output. If you have any issues with the file upload, please let me know. <https://www.nature.com/documents/nr-software-policy.pdf>

* Please confirm in your cover letter whether your study is compliant with the "Guidance of the Ministry of Science and Technology (MOST) for the Review and Approval of Human Genetic Resources", which requires formal approval for the export of human genetic material or data from China.

DATA AND CODE AVAILABILITY (This seems to be just general things)

* All Nature Communications manuscripts must include a “Data Availability” section after the Methods section but before the References. If any of the data can only be shared on request or are subject to restrictions, please specify the reasons and explain how, when, and by whom the data can be accessed. For more information on this policy and a list of examples, see:

<https://www.nature.com/documents/nr-data-availability-statements-data-citations.pdf>

* Please also include a “Code Availability” section after the “Data Availability” section. If the code can only be shared on request, please specify the reasons. For more information on our code sharing policy and requirements, please see:

* All novel microarray, DNA sequencing, RNA-seq or proteomic datasets must be deposited in a publicly accessible database, and accession codes and associated hyperlinks must be provided in the “Data Availability” section.

* To maximise the reproducibility of research data, we strongly encourage you to provide a file containing the raw data underlying the following types of display items:

- Any reported means/averages in box plots, bar charts, and tables
- Dot plots/scatter plots, especially when there are overlapping points
- Line graphs

The data should be provided in a single Excel file with data for each figure/table in a separate sheet, or in multiple labelled files within a zipped folder. Name this file or folder 'Source Data', and include a brief description in your cover letter. The "Data Availability" section should also include the statement "Source data are provided with this paper."

To learn more about our motivation behind this policy, please see: <https://www.nature.com/articles/s41467-018-06012-8>

* Please replace your bar graphs with plots that feature information about the distribution of the underlying data. All data points should be shown for plots with a sample size less than 10. For larger sample sizes, please consider box-and-whisker or violin plots as alternatives. Measures of centrality, dispersion and/or error bars should be plotted and described in the figure legend.

REVIEWERS' COMMENTS

Reviewer #1 (Remarks to the Author):

In this revised manuscript, Lee et al explore the hypothesis that country-wide antibiotic usage patterns correlate with antibiotic resistance observed in both pathogens and commensals residing in human microbiomes. The majority of the study is focused on the stool microbiome. They find a correlation between antibiotic use and ARG abundance, identify two 'resistotypes', and also assess the relative contribution of MGE-derived ARGs to their observations. They postulate that ARGs are being transferred between commensals and pathogens, and cite this as an important reason for evaluating ARGs in microbiomes. The authors were highly responsive to reviewer comments. The MGE analysis is much deeper and improved compared to prior. In general, the most interesting finding to me is the relationship between microbiome ARG abundance (measured two different ways) and community-wide antibiotic use. Overall, the manuscript is timely and the bioinformatic experiments are well carried out. There are some typographical errors throughout the manuscript that could use some attention. I have a few major and several minor comments that are enumerated below, that if addressed would enhance the clarity and potential impact of the manuscript.

Major comments:

1. I wonder if the less frequent resistotype is simply a lower biomass sample where absolute abundance of bacteria has decreased due to antibiotic-mediated clearing of commensals. There is existing and emerging literature on absolute abundance and its utility in microbiome research – e.g. <https://www.nature.com/articles/nature24460> and <https://www.biorxiv.org/content/10.1101/2022.09.28.509972v2>) – might be worth discussing in the discussion.
2. The authors focus on the high rate of AMR in China; however, in figure 2a, the read-based approach to identifying AMR suggests that the number of AMR genes per genome is much higher in Spain and France than in China. What do the authors make of this?
3. I think the finding that France and the Netherlands have AMR that is on opposite sides of the spectrum based on the cpg metric is very interesting – especially since this seems to correlate well with antibiotic usage differences in these countries that are rather close to one another.

Minor comments:

1. Abstract: “The less frequent resistotype, has” – no comma need between 'resistotype' and 'has'
2. Line 5 – “citeDeKraker2016” should probably be cleaned up.

3. Line 24 – considering that we don't know whether proteobacteria are indeed increasing or “blooming” or commensals are decreasing and proteobacteria are staying the same (or also decreasing in concentration albeit to a lesser degree), it may be prudent to avoid the term “blooming” and rather say something that makes it clear that this is being evaluated based on compositional data.
4. Line 103 – can some detail be provided on what “airway” samples are? There are several types of samples that I would consider “airway”, and they are rather different from one another (e.g. sputum, BAL, nares swabs)
5. Line 104 still mentions a milk sample – but I thought this was excluded in this revision based on the response to reviewers comments document?
6. Line 113-114 – the authors identified a median of 14 ARGs per metagenome – this number seems low to me given how diverse their major contributor to this dataset (stool) is. Also, it might make sense to report median and range of ARGs per body type – as I'd expect it to be very different in skin vs. stool or oral cavity.
7. Line 149ff – what is being calculated and reported in the ARG ‘diversity’ metric? Based on the results text, it is not clear. Is this richness (and therefore count)? I presume not because the median ARGs is lower than the metric reported here.
8. Lines 152-153 – would be consistent with number of significant digits that are used.
9. Line 177 – the sentence starting with “Implying” is somewhat awkwardly written.
10. Line 180 – might reword this sentence to avoid the use of the word “sick”. Perhaps “healthy and those who have at least one diagnosed disease”... or something like that.
11. Regarding “healthy controls” – it is always a challenge to ascertain what “healthy” means in these microbiome studies because often there is no strict enrollment criteria for these studies. You might like to point out that limitation when this is first brought up. These are individuals who self-identify as healthy, and the inclusion/exclusion criteria for what is considered healthy varies by study. I do think a strength of this study is that the final “healthy” control population that was investigated were only those who were unambiguously identified as not taking antibiotics at the time of sample collection.
12. In the Figure 2 legend, please provide a ‘key’ for the country abbreviations that are used.
13. Line 310 – integrases is misspelled.
14. Line 468 – the title “country level response” could be a bit more specific so the reader knows from the title what the “response” is to.
15. Line 495 – need a comma between ‘diseases’ and ‘particularly’
16. Line 513 – it is surprising to me that the FAMP resistotype isn't correlated with abx consumption rate – this is worthy of discussion.

17. Figure 6 – while the cited study (10) did use oral meropenem, unless I am mistaken, meropenem is not typically used orally. This should be discussed.

<https://www.sciencedirect.com/science/article/abs/pii/S1359644620304694>

<https://go.drugbank.com/drugs/DB00760>

18. Line 587- using the term “ascribed” might conflate association with causation. Consider rewording.

19. Line 590 – can you be more precise regarding what “higher levels of resistance” means? It can mean a variety of things (higher MIC, more organisms being resistant, organisms being resistant to a larger number of abx, etc.)

20. I’m surprised that beta lactams and macrolides are so much more commonly used than fluoroquinolones – it would be interesting to know the rate of FQ use as well (in the discussion, worth citing that number).

21. Discussion – I think China, France and Spain are on one end of the spectrum and all are worthy of discussion. Then you can jump into the discussion of China as an outlier with respect to ARG abundance being high but a relatively low reported rate of antibiotic consumption. This might enrich the discussion and also make the findings more relevant for Europeans. As noted above, I find the contrast between the Netherlands and France to be particularly interesting. Another note on the interesting ‘outlier’ of China – might this be due to use of other medications? For example, the labs of Typas and Bork demonstrated that many medications (not just antibiotics) have strong antimicrobial effects on commensals. <https://www.nature.com/articles/nature25979>

22. A small point, but in the answer to reviewer 1, question 5 - it seems that upper limb is missing from the two graphs on the right (x axis - subsamples N).

23. Answer to reviewer 1, minor comment 9 - I find it interesting that there is a lower abundance of SGBs in samples from China compared to other countries. Is this expected? probably out of the scope of this manuscript to dig into lots of detail on this, but this may be what is driving the high ARG/SGB statistic.

Reviewer #2 (Remarks to the Author):

The authors accepted most of my suggestions and have substantially revised their analysis and paper, which was great to see. I also believe that the discussion of the results is much more balanced in the revised manuscript. I have no major concerns remaining.

Minor comments:

Please check these two papers that I believe have reported lack of correlation between antibiotic usage and antibiotic resistance gene abundance (I agree that the issue is complicated and antibiotic usage is not documented well at all. The authors have done a great effort on this, it seems)

<https://doi.org/10.1021/acs.est.1c08673>

<https://doi.org/10.1038/s41467-019-08853-3>

(in the supplementary material)

I don't think it is appropriate to cite reference #31 for the 95% ANI threshold for species but rather the work of Konstantinidis and colleagues. The latest on that topic and hence, a good reference to cite is a Nat. Comms. paper, Rodriguez-R et al., 2021, I think.

“in most cases we longer found correlations with the notable exception of the Beta-lactamas where for abundance we did observe a significant

correlations for the WHO data”. Is “no” missing in front of longer? I also believe it should be written as “lactams” not lactamas.

Reviewer #3 (Remarks to the Author):

In this manuscript, the authors conducted a large survey of antibiotic resistance genes (ARGs) in publicly available metagenome assemblies spanning multiple countries. The authors claim that the ARG richness in metagenomes positively correlates with the reported antibiotic usage rates in the countries under investigation. Most of the comments from the review of the first submission were addressed by the authors. However, I believe the following outstanding points need to be addressed before the manuscript is considered for publication.

Major comments:

1. As pointed out in the original review, strain pathogenicity is a highly complex trait and has high within-species variability depending on several factors. The authors' further work to base the pathogenicity on virulence factors is appreciated; however, the general definition of virulence

factors is vast and very vague. As such, I still insist that classification of SGBs as pathogenic or non-pathogenic depending primarily on previous reports of infection at any site in human body, even when bolstered by enumeration of virulence factors, is inappropriate and misleading. Claims regarding pathogenicity of SGBs need to be revised, and the authors need to tone down the language in the corresponding sections.

2. How did the qualities of the MAGs of origin compare across ARGs pertaining to different antibiotic classes? If the MAG quality differs significantly and substantially, that would likely affect the number of instances ARGs from a given antibiotic class are identified, skewing the results reported in the “ARG novelty varied with respect to antibiotic classes” section. The authors need to look into this and address it depending on the MAG quality comparative analysis. Similarly, the authors should compare the MAG qualities across body sites in the “ARG diversity varied across body sites” section.

Minor comments:

Line 5: Remove “citeDeKraker2016” and cite the mentioned paper.

Lines 6-7: The statement “to date, the majority of AMR surveillance consists of resistance rates in pathogen isolates cultured from infections”...

“Proteobacteria” is not italicized.

Line 508: It is advised to change “genders” to “sexes.”

Line 601: change “phenomena” to “phenomenon”

Line 638: change “phenomenon” to “phenomena”

Response to reviewers

Reviewer 1

In this revised manuscript, Lee et al explore the hypothesis that country-wide antibiotic usage patterns correlate with antibiotic resistance observed in both pathogens and commensals residing in human microbiomes. The majority of the study is focused on the stool microbiome. They find a correlation between antibiotic use and ARG abundance, identify two 'resistotypes', and also assess the relative contribution of MGE-derived ARGs to their observations. They postulate that ARGs are being transferred between commensals and pathogens, and cite this as an important reason for evaluating ARGs in microbiomes. The authors were highly responsive to reviewer comments. The MGE analysis is much deeper and improved compared to prior. In general, the most interesting finding to me is the relationship between microbiome ARG abundance (measured two different ways) and community-wide antibiotic use. Overall, the manuscript is timely and the bioinformatic experiments are well carried out. There are some typographical errors throughout the manuscript that could use some attention. I have a few major and several minor comments that are enumerated below, that if addressed would enhance the clarity and potential impact of the manuscript.

Major comments:

I wonder if the less frequent resistotype is simply a lower biomass sample where absolute abundance of bacteria has decreased due to antibiotic-mediated clearing of commensals. There is existing and emerging literature on absolute abundance and its utility in microbiome research – e.g. <https://www.nature.com/articles/nature24460> and <https://www.biorxiv.org/content/10.1101/2022.09.28.509972v2>) – might be worth discussing in the discussion.

We agree it is important to bear in mind the limitation of metagenomics to changes in relative abundance so we have as the reviewer suggests added some discussion of this see Lines 690-694:

“Secondly, metagenomics can only determine relative changes in abundance so for example in the FAMP resistotype it is possible that the absolute abundance of resistant pathogens is not higher, rather that the susceptible commensals have decreased, this would motivate revisiting these observations with methods for quantifying absolute microbial loads [49].”

2. The authors focus on the high rate of AMR in China; however, in figure 2a, the read-based approach to identifying AMR suggests that the number of AMR genes per genome is much higher in Spain and France than in China. What do the authors make of this?

This is correct, China has the third highest normalized abundance of ARGs with France and Spain at ranks two and one respectively having higher ARG levels. The key point though is that China is a major outlier with respect to the otherwise linear association between ARG abundance and antibiotic consumption. China has the lowest reported consumption rate whereas Spain and France have the highest reported consumption rates in our group of countries. China has far higher resistance levels than we would expect, France and Spain the level we would expect. That is why we emphasise the unexpected levels of resistance in that country.

3. I think the finding that France and the Netherlands have AMR that is on opposite sides of the spectrum based on the cpg metric is very interesting – especially since this seems to correlate well with antibiotic usage differences in these countries that are rather close to one another.

We agree and we have added discussion of this point (see lines 638-641).

Minor comments:

1. Abstract: “The less frequent resistotype, has” – no comma need between ‘resistotype ’and ‘has’
Changed

2. Line 5 “ –citeDeKraker2016” should probably be cleaned up.
Changed

3. Line 24 – considering that we don’t know whether proteobacteria are indeed increasing or “blooming” or commensals are decreasing and proteobacteria are staying the same (or also decreasing in concentration albeit to a lesser degree), it may be prudent to avoid the term “blooming” and rather say something that makes it clear that this is being evaluated based on compositional data.

Changed ‘blooming’ to a ‘community dominated by’.

4. Line 103 – can some detail be provided on what “airway” samples are? There are several types of samples that I would consider “airway”, and they are rather different from one another (e.g. sputum, BAL, nares swabs)

These were in fact sputum samples and we have now clarified that in the text adding this sentence:

‘More specifically, sample types classified as ‘oral cavity’ include samples from plaque (222), tongue (189), buccal mucosa (118), others or unspecified (217); ‘nasal cavity’ includes anterior nares (55); ‘airway’ corresponds to sputum (118).’

5. Line 104 still mentions a milk sample – but I thought this was excluded in this revision based on the response to reviewers comments document?

Corrected.

6. Line 113-114 – the authors identified a median of 14 ARGs per metagenome – this number seems low to me given how diverse their major contributor to this dataset (stool) is. Also, it might make sense to report median and range of ARGs per body type – as I’d expect it to be very different in skin vs. stool or oral cavity.

We agree it is quite low, it is important to bear in mind though that this is the number of assembled ARGs found and a number of factors will influence the rate of recovery most notably sequencing depth. As the reviewer suspected though it does vary from body to body site and we have now added these numbers to the manuscript. We do not want to emphasise these too much though as sequencing depth will vary across these samples, preferring instead to focus on the corrected rarefied read based diversities that we present in the next section. Note that in fact the correct median across all sites is actually 15 not 14, the earlier value applied to a very slightly different data set and we apologise for the oversight.

7. Line 149ff – what is being calculated and reported in the ARG ‘diversity’ metric? Based on the results text, it is not clear. Is this richness (and therefore count)? I presume not because the median ARGs is lower than the metric reported here.

It is the rarefied number of ARGs and we have clarified that in the text. The number reported here 9.245 is different from the 15 mentioned above both because these are the read-based results and because they have been rarefied to correct for different sequencing depths across samples.

8. Lines 152-153 – would be consistent with number of significant digits that are used.

We have used the same number of significant digits throughout now.

9. Line 177 – the sentence starting with “Implying” is somewhat awkwardly written.

We have reworded this sentence.

10. Line 180 – might reword this sentence to avoid the use of the word “sick”. Perhaps “healthy and those who have at least one diagnosed disease”... or something like that.

We have changed this sentence.

11. Regarding “healthy controls” – it is always a challenge to ascertain what “healthy” means in these microbiome studies because often there is no strict enrollment criteria for these studies. You might like to point out that limitation when this is first brought up. These are individuals who self-identify as healthy, and the inclusion/exclusion criteria for what is considered healthy varies by study. I do think a strength of this study is that the final “healthy” control population that was investigated were only those who were unambiguously identified as not taking antibiotics at the time of sample collection.

We agree and we have added the following caveat when we first mention healthy controls:

‘It is important to note that because this is a meta-analysis there is no single definition of healthy control which might vary from one study to another.’

12. In the Figure 2 legend, please provide a ‘key ’for the country abbreviations that are used.

These have now been added.

13. Line 310 – integrases is misspelled.

Corrected

14. Line 468 – the title “country level response” could be a bit more specific so the reader knows from the title what the “response” is to.

We have changed this to ‘Country-level response to antibiotic consumption’.

15. Line 495 – need a comma between ‘diseases ’and ‘particularly’

Changed.

16. Line 513 – it is surprising to me that the FAMP resistotype isn’t correlated with abx consumption rate – this is worthy of discussion.

This was surprising to us too and we do refer to this already in the Discussion. In fact, since short term consumption is associated with the FAMP resistotype it is potentially evidence that we have successfully screened individuals that have recently consumed antibiotics.

17. Figure 6 – while the cited study (10) did use oral meropenem, unless I am mistaken, meropenem is not typically used orally. This should be discussed.

<https://www.sciencedirect.com/science/article/abs/pii/S1359644620304694>

<https://go.drugbank.com/drugs/DB00760>

We have now clarified the antibiotic regime used in the Palleja study:

‘In this study twelve men received a cocktail of three last-resort antibiotics orally: meropenem, gentamicin and vancomycin for four days and their gut microbiome tracked for six months.’

18. Line 587- using the term “ascribed” might conflate association with causation. Consider rewording.

We have changed ‘ascribed’ to ‘explained by’.

19. Line 590 – can you be more precise regarding what “higher levels of resistance” means? It can mean a variety of things (higher MIC, more organisms being resistant, organisms being resistant to a larger number of abx, etc.)

We have now clarified this to: ‘Previous studies have shown a higher abundance of resistance genes in microbiomes from individuals deriving from countries with higher antibiotic consumption’.

20. I’m surprised that beta lactams and macrolides are so much more commonly used than fluoroquinolones – it would be interesting to know the rate of FQ use as well (in the discussion, worth citing that number).

We have now added the mean percentage fluoroquinolone usage statistics which were a mean of 10.0% from the WHO data slightly less than the macrolides.

21. Discussion – I think China, France and Spain are on one end of the spectrum and all are worthy of discussion. Then you can jump into the discussion of China as an outlier with respect to ARG abundance being high but a relatively low reported rate of antibiotic consumption. This might enrich the discussion and also make the findings more relevant for Europeans. As noted above, I find the contrast between the Netherlands and France to be particularly interesting. Another note on the interesting ‘outlier’ of China – might this be due to use of other medications? For example, the labs of Typas and Bork demonstrated that many medications (not just antibiotics) have strong antimicrobial effects on commensals. <https://www.nature.com/articles/nature25979>

We have now added discussion of Spain and France in contrast to the Netherlands. It could be due to other medications but given an absence of any statistics to confirm this and given that we know agricultural usage of antibiotics is high in China we prefer that as a speculative hypothesis.

22. A small point, but in the answer to reviewer 1, question 5 - it seems that upper limb is missing from the two graphs on the right (x axis - subsamples N).

The different sites have different sample numbers so that the curves have different lengths.

23. Answer to reviewer 1, minor comment 9 - I find it interesting that there is a lower abundance of SGBs in samples from China compared to other countries. Is this expected? probably out of the scope of this manuscript to dig into lots of detail on this, but this may be what is driving the high ARG/SGB statistic.

This is an interesting observation but these were uncorrected for sampling depth which may be the explanation and the effect is not large enough (a factor of two) to explain the anomalously high levels in China which are many times larger than we would expect given the antibiotic consumption rate.

Reviewer #2 (Remarks to the Author):

The authors accepted most of my suggestions and have substantially revised their analysis and paper, which was great to see. I also believe that the discussion of the results is much more balanced in the revised manuscript. I have no major concerns remaining.

Minor comments:

Please check these two papers that I believe have reported lack of correlation between antibiotic usage and antibiotic resistance gene abundance (I agree that the issue is complicated and

antibiotic usage is not documented well at all. The authors have done a great effort on this, it seems)

<https://doi.org/10.1021/acs.est.1c08673>

<https://doi.org/10.1038/s41467-019-08853-3>

(in the supplementary material)

We thank the reviewer for bringing these studies to our attention and we have added some discussion of this point to the Discussion where we cite them:

‘This is in contrast to global wastewater metagenome surveys of ARGs which have so far failed to find a clear link between antibiotic consumption and ARG abundance [44, 45]. This highlights the importance of sampling microbiomes directly rather than from waste streams where additional factors may be impacting abundance.’

I don't think it is appropriate to cite reference #31 for the 95% ANI threshold for species but rather the work of Konstantinidis and colleagues. The latest on that topic and hence, a good reference to cite is a Nat. Comms. paper, Rodriguez-R et al., 2021, I think.

We agree and we have changed the relevant citation to that suggested by the reviewer.

“in most cases we longer found correlations with the notable exception of the Beta-lactamas where for abundance we did observe a significant correlations for the WHO data”. Is “no” missing in front of longer? I also believe it should be written as “lactams” not lactamas.

Thanks, yes we have corrected both these errors.

Reviewer #3 (Remarks to the Author):

In this manuscript, the authors conducted a large survey of antibiotic resistance genes (ARGs) in publicly available metagenome assemblies spanning multiple countries. The authors claim that the ARG richness in metagenomes positively correlates with the reported antibiotic usage rates in the countries under investigation. Most of the comments from the review of the first submission were addressed by the authors. However, I believe the following outstanding points need to be addressed before the manuscript is considered for publication.

Major comments:

1. As pointed out in the original review, strain pathogenicity is a highly complex trait and has high within-species variability depending on several factors. The authors' further work to base the pathogenicity on virulence factors is appreciated; however, the general definition of virulence factors is vast and very vague. As such, I still insist that classification of SGBs as pathogenic or non-pathogenic depending primarily on previous reports of infection at any site in human body, even when bolstered by enumeration of virulence factors, is inappropriate and misleading. Claims regarding pathogenicity of SGBs need to be revised, and the authors need to tone down the language in the corresponding sections.

We are very clear how we define pathogenicity and we discuss its limitations when we introduce the definition. We agree with the reviewer that there is no perfect way to do this but the fact that we do see a good correlation between our method and virulence factor occurrence is reassuring. Furthermore, for the type of broad statistics that we apply it to, it will not matter if the definition is not perfect, it just needs to be correct enough of the time to tease apart difference between pathogens and non-pathogens. The fact that we see such differences e.g. the FAMP resistotype being associated with pathogenic species and the country level response being associated with commensals suggests that our definition is good enough in the context that we apply it. To reiterate this point though we have added a further caveat to the Discussion section:

‘We should add two important caveats to the above conclusions, firstly our definition of a pathogenic species, as one with a strain reported to have caused infection at any body site, is imperfect and probably overly broad. Therefore the ARGs may actually be carried on non-

pathogenic strains of opportunistic pathogen species. However, the associations we observe suggest that in a statistical sense our definition is useful and no better definition was apparent to us.'

2. How did the qualities of the MAGs of origin compare across ARGs pertaining to different antibiotic classes? If the MAG quality differs significantly and substantially, that would likely affect the number of instances ARGs from a given antibiotic class are identified, skewing the results reported in the "ARG novelty varied with respect to antibiotic classes" section. The authors need to look into this and address it depending on the MAG quality comparative analysis. Similarly, the authors should compare the MAG qualities across body sites in the "ARG diversity varied across body sites" section.

The reviewer is under a misconception here, the ARG collection is generated through annotation of de novo assemblies from individual samples, it is not dependent on MAG construction. MAG construction is only used to assign taxonomy to ARGs and hence identify mobile or species-specific clusters. The reviewer's concerns regarding the impact of MAG quality on ARG novelty etc. are unfounded and, hence, this additional analysis unnecessary.

Minor comments:

Line 5: Remove "citeDeKraker2016" and cite the mentioned paper.

Changed.

Lines 6-7: The statement "to date, the majority of AMR surveillance consists of resistance rates in pathogen isolates cultured from infections"...

Clarified

"Proteobacteria" is not italicized.

Changed

Line 508: It is advised to change "genders" to "sexes."

Line 601: change "phenomena" to "phenomenon"

Line 638: change "phenomenon" to "phenomena"

All changed.